# INTENTION MODEL: A NOVEL EXPLANATION FOR IN-CONTEXT LEARNING

## ABSTRACT

In-context learning (ICL) has demonstrated remarkable success in enabling large language models (LLMs) to learn to do a downstream task by simply conditioning on a few input-output demonstrations. Distinct from traditional learning paradigms, ICL does not require model updates, thus attracting significant interest in understanding the mechanisms behind LLMs' ICL capabilities. Advanced works aim to understand ICL through an empirical viewpoint to provide the multifaceted nature of ICL, while some works aim to explain how ICL can emerge theoretically. However, the current theoretical analysis exhibits a weak connection to empirical explorations due to strong assumptions, e.g., perfect LLMs and ideal demonstrations. This work proposes an intention model, providing a novel theoretical framework for explaining ICL. With mild assumptions, we present a "no-free-lunch" theorem for ICL: whether ICL emerges depends on the prediction error and prediction noise, which are determined by *i)* LLMs' error of next-token prediction, *ii)* LLMs' prediction smoothness, and *iii)* the quality of demonstrations. Moreover, our intention model provides a novel explanation for the learning behavior of ICL under various input-output relations, e.g., learning with flipped labels. This is fortunately consistent with our experimental observations.

## 1 INTRODUCTION

In-context learning (ICL) refers to the capability of large language models (LLMs) to adapt to new tasks, which is achieved through conditioning on a few input-output demonstrations without model updates (Brown et al., 2020). It shows that the striking feature of LLMs is inherently related to the model scale (Brown et al., 2020; Garg et al., 2022; Wei et al., 2023) and demonstrations, like the selected pairs, order of pairs, and the format of demonstrations (Min et al., 2022; Liu et al., 2022; Wang et al., 2024). According to the understanding of traditional learning paradigms, it is difficult to anticipate LLMs' ICL capacity because they are trained to predict the next token rather than perform like in-context predictors (Ouyang et al., 2022). This intriguing mismatch between ICL and traditional learning paradigms has attracted increasing attention in understanding LLMs' ICL capability (Dong et al., 2023).

Numerous outstanding works have revealed the enigmatic characteristics inherent to ICL. Specifically, it shows that leveraging randomly assigned labels of the inputs leads to a limited change in performance (Min et al., 2022) when the number of input-output pairs is limited (Kossen et al., 2024), challenging the necessary of ground-truth labels in demonstrations. Meanwhile, it shows that the performance change induced by flipping labels of input demonstrations depends on the model scale (Wei et al., 2023). Namely, LLMs with larger model scale can override the semantic priors (i.e., predict "positive" for samples from the positive class) by predicting the flipped label (i.e., predict "negative" for samples from the positive class), where demonstrations are composed of inputs with flipped labels (assigning positive labels for inputs from the negative class). These intriguing phenomena highlight the difference between traditional learning and ICL (Kossen et al., 2024). These seminal explorations provide a fruitful guide for understanding and explaining ICL.

In addition to empirical investigations, recent advancements have yielded significant theoretical insights into the mechanisms underpinning the ICL capabilities of LLMs. For instance, the seminal work points out that ICL emerges because LLMs explicitly perform Bayesian inference (Xie et al., 2022), inspiring numerous explorations (Hahn & Goyal, 2023; Jiang, 2023). Meanwhile,

Von Oswald et al. (2023); Akyürek et al. (2023) argue that a traditional gradient decent approach can endow transformer-based models with the ICL capability, shedding light to identify model behaviors (Reddy, 2024). However, these studies often demonstrate a relatively weak connection to empirical investigations, potentially due to strong assumptions such as the perfect alignment of LLMs with data distributions and the ideal representation of tasks by demonstrations.

Inspired by Brown et al. (2020); Xie et al. (2022), we propose a novel framework to theoretically explain LLMs' ICL capability, sharing the same spirit as the topic model (Blei et al., 2003). According to the intention model, *LLMs can infer intentions from demonstrations by filtering out unrelated intentions (Eq. 5). Using inferred intentions, LLMs can predict ground-truth outputs with bounded prediction errors (Eq. 7) and noise (Eqs. 8, and 9).* Our **main contributions** are twofold. First, we propose a no-free-lunch Theorem 1 for ICL to highlight that the conditions for the emergence of ICL are not naturally met. Second, our intention model takes a step towards bridging the gap between theoretical and empirical results by providing a novel explanation for the learning behavior of ICL when LLMs are conditioned on varying input-output mappings.

First, whether ICL emerges depends on the prediction error and prediction noise, which are determined by *i)* LLMs' error of next-token prediction, *ii)* LLMs' prediction smoothness, and *iii)* the quality of demonstrations. This aligns with the understanding of ICL and highlights the necessity of a small error in predicting the next token for ICL. These theoretical results underscore the contribution of our intention model, marking the first instance of relaxing the strong assumptions regarding: *i)* perfect alignment of LLMs, and *ii)* the idealized representation of tasks by demonstrations. Second, the scenarios of ICL with varying input-output mappings can be modeled through an external transition matrix (Eq. 10), leading to the following conclusions.

- Introducing an external mapping to modify the original outputs of demonstrations makes ICL more challenging. This is because the transition matrix would increase the upper bound of prediction noise (Eq. 9). This is consistent with our results shown in Table 1.
- An LLM with a smaller error of next-token prediction performs better in overriding semantic priors under flipped label scenarios. This is because the smaller error can lead to a reduced prediction noise (Eq. 9). This conclusion aligns well with our experimental observations (Table 1) and previous experimental observations (Wei et al., 2023).
- Increasing the number of demonstrations under the random label scenario decreases performance. This is because a larger number of demonstrations would magnify the impact of demonstration shift and the LLMs' error of next-token prediction, as shown in Eq. 9. This conclusion aligns well with previous experimental observations (Kossen et al., 2024).

## 2 PRELIMINARY

In-context learning (ICL) is firstly introduced by the seminal work (Brown et al., 2020), where LLMs are conditioned on a few demonstrations of a task and can predict what comes next to complete the task. Although studying the special ICL can reveal the difference between traditional learning and ICL (Reddy, 2024), this paper mainly focuses on exploring what enables LLMs' ICL ability following previous works (Brown et al., 2020; Xie et al., 2022). Thus, we mainly study the general ICL ability of LLMs following Brown et al. (2020), which is defined as follows.

**Definition 1 (In-context learning)** *In-context learning refers to the ability with which LLMs can complete a task simply by conditioning on a few input-output demonstrations of the task.*

According to Definition 1, ICL introduces a novel task adaptation approach that contrasts with updating model parameters in the traditional learning paradigm. This is achieved through task demonstrations, as mentioned in the definition.

**Demonstration structure.** The task demonstrations for ICL are constructed by natural language instruction composed of input-output pairs $O_i = \{(x_i, y_i)\}$, which are constructed to form the representation of tasks (Brown et al., 2020). A test sample $x_t$ is typically concatenated at the end of these input-output pairs. Thus, the demonstration structure of ICL is, $[S^n, x_t] = [x_1, y_1, o_1^f, x_2, y_2, o_2^f, \cdots, x_n, y_n, o_n^f, x_t]$, where $n$ is the number of input-output pairs, and $o_i^f$ presents the format in ICL, e.g., delimiter tokens.

**Demonstration construction.** Given a text sample $x_t$, the demonstration of ICL is constructed for the corresponding task descriptions. Based on the constructed demonstrations, the in-context predictor $f_M(S^n, x_t)$ outputs the prediction through a LLM $M$ with the highest probability of $x_t$:

$$f_M(S^n, x_t) = \arg\max_y p_M(y|S^n, x_t), \tag{1}$$

where $y$ represents a set of tokens from the vocabulary set $\mathcal{O}$ and $p_M(\cdot|\cdot)$ stands for the prediction probability of an LLM. In this context, the in-context predictor is expected to minimize the prediction risk by constructing appropriate demonstrations $S^n$: $\min_{S^n} \mathbb{E}_{(x_t, y_t)} \mathbb{I}(y_t \neq f_M(S^n, x_t))$, where $y_t$ is the ground-truth output of the test input $x_t$ and $\mathbb{I}(\cdot)$ represents the indicator function. Recent works show that tuning the format, selecting different input-output pairs, and changing the order of input-output pairs in the demonstrations can significantly affect ICL performance.

Searching for demonstrations to promote ICL performance is similar to prompt learning (Shin et al., 2020; Zhou et al., 2022), i.e., tuning prompts to adapt large-scale models to downstream tasks. However, the mechanism involved in ICL could be beyond the feature alignment between modalities (Brown et al., 2020). Moreover, optimizing prompts in the embedding space is challenging for ICL due to the large and discrete search space (Zou et al., 2023). Thus, the current methods mainly focus on hand-crafted demonstrations. In this context, ICL performance heavily relies on two factors: i) the construction of demonstrations and ii) the performance of LLMs.

## 3 SETUP OF INTENTION MODEL

Our intention model for explaining ICL of LLMs is mainly inspired by the seminal works (Blei et al., 2003; Brown et al., 2020; Xie et al., 2022).

### 3.1 FORMALIZING ICL WITH INTENTION MODEL

Let all tokens come from the vocabulary set $\mathcal{O}$, and tokens $o \in \mathcal{O}$ form the basic unit of text. Consider the process of generating a document, $\mathcal{D}^k = \{o_1, \cdots, o_k\}$, with $k$ observed tokens. Our intention model assumes that the process starts by determining a description intention $\theta \in \Theta$. The intention reflects the semantics and context of the text. According to the intention $\theta$, words or tokens are generated one by one until the document is completed. Namely, the distribution of a document can be formalized as, $q(\mathcal{D}^k) = q(o_1, \cdots, o_k) = \int_\Theta q(o_1, \cdots, o_k|\theta)q(\theta)d\theta$, where the intention $\theta$ determines the generation process of a document $\mathcal{D}^k$. Given a test input $x_t = \mathcal{D}_x^k$, demonstrations are constructed to describe the task, determining the expected output of $x_t$, i.e., $y_t = \mathcal{D}_y^k$. According to the text generation, demonstrations $S^n$ can be constructed by a set of input-output pairs $(x, y)$ generated with a demonstration intention $\theta_d$, $S^n(\theta_d) = \{O_i(\theta_d), o_i^f\}_{i=1}^n = \{x_i, y_i, o_i^f\}_{i=1}^n$, with $O_i(\theta_d) \triangleq (x_i, y_i) \sim q(x, y|\theta_d)$. In practical scenarios, the demonstration intention $\theta_d$ used to construct demonstrations is controlled by users, where the distribution of delimiters is independent of intentions because task intentions do not control the delimiter.

The selection of distinct input-output pairs for task demonstration yields varying ICL performance. This is inherently related to the estimation error of the task's ground-truth intention $\theta_g$, i.e., $\theta_d \neq \theta_g$. It is necessary to highlight the fact that the output $y_t(\theta_g)$ for the same input $x_t$ would vary with the ground-truth intention $\theta_g$, e.g., outputs are flipped under flipping label scenarios.

Considering the discrepancy between the demonstration intention $\theta_d$ and the ground-truth intention $\theta_g$, we employ the Kullback-Leibler divergence[1] to capture this discrepancy: $\mathrm{KL}\left(q\left(x, y|\theta_d\right) \| q\left(x, y|\theta_g\right)\right) < \epsilon$, where $\epsilon$ captures the demonstration shift, i.e., high-quality demonstrations lead to a small $\epsilon$. We introduce a neighborhood $\Theta_\epsilon(\theta_g)$ for $\theta_g$: $\forall \theta \in \Theta_\epsilon(\theta_g), \mathrm{KL}\left(q\left(x, y|\theta\right) \| q\left(x, y|\theta_g\right)\right) < \epsilon$. We use $\Theta_\epsilon$ to denote the neighborhood of $\theta_g$. Note that this highlights the difference between our work and a previous work (Xie et al., 2022), where demonstrations can ideally represent the task, i.e., $\theta_d = \theta_g$.

In the context of the intention model, the prediction of an LLM can be formalized as follows,

$$p_M(y|S^n(\theta_d), x_t) = \int_\Theta p_M(y|S^n(\theta_d), x_t, \theta)p_M(\theta|S^n(\theta_d), x_t)d\theta. \tag{2}$$

---

[1]Other metrics can also be employed to model the distribution discrepancy, which is left as our future work.

Here, the first term $p_M(y|S^n(\theta_d), x_t, \theta)$ represents the generated outputs given the intention $\theta$, an input query $x_t$, and the demonstrations $S^n(\theta_d)$. We do not introduce the assumption that $p_M(y|S^n(\theta_d), x_t, \theta) = p_M(y|x_t, \theta)$ according to the empirical explorations (Wei et al., 2023). Namely, the outputs could be sampled from a semantically unrelated label space, e.g., assigning *bar/foo* as the label of *positive/negative*, implying that the output relies heavily on the demonstrations $S^n(\theta_d)$. Thus, the first term shares the same spirit with the generation process of a topic model (Blei et al., 2003). Meanwhile, the second term shows the same spirit as a topic model, where the intention/topic can be generated from the context.

The intuition of Eq. 2 is straightforward. Specifically, there could be numerous possible outputs $p_M(y|S^n(\theta_d), x_t, \theta)$ varying with the given demonstrations $S^n(\theta_d)$, input query $x_t$, and the intention $\theta$. All these outputs could be the output of an in-context predictor. In this context, given a demonstration intention $\theta_d$ used to construct demonstrations, $p_M(\theta|S^n(\theta_d), x_t)$ works like a filter to "select" intention(s) such that the output can match the ground-truth intention $\theta_g$.

### 3.2 INTENTION MODEL REALIZATION

Inspired by Xie et al. (2022), we leverage the Hidden Markov Model (HMM) to model the process of one-by-one word generation. In this context, the intention $\theta$ parameterizes the transition probability matrix of HMM hidden states $h_1, \cdots, h_k \in \mathcal{H}$, while the hidden state $h_i$ parameterizes the distribution of the token $o_i$. Namely, the hidden state space $\mathcal{H}$ contains hidden states used for text generation that control the text generation. Then, we can re-write the generation process of a test output as follows,

$$y_t(\theta_g) = \arg\max_y p(y|x_t, \theta_g) \text{ with } x_t \sim q(x|h'_t, \theta_g), \tag{3}$$

where $h'_t$ represents the start hidden state of the test input $x_t$. Besides the input-output pairs, formats play a crucial role in ICL, which are typically realized with delimiters, e.g., "\n". A delimiter $o^f$ denoting the format in ICL can be sampled from the distribution $q(o^f|h^f)$, where $h^f$ is the hidden state sampled from a subset $\mathcal{H}^f$ of state space $\mathcal{H}$, i.e., $h^f \in \mathcal{H}^f \subseteq \mathcal{H}$. Then, the demonstrations with $n$ independently sampled input-output pairs can be formalized with $S_i^n(\theta_d)$ the $i$-th pair as follows,

$$p\left(S^n(\theta_d)\right) = \prod_{i=2}^n \sum_{h_i^f \in \mathcal{H}^f} p\left(S_i^n(\theta_d)|h_{i-1}^f, \theta_d\right) p\left(h_{i-1}^f|S_{i-1}^n(\theta_d), \theta_d\right). \tag{4}$$

The text is determined by a given intention $\theta$ and the start state $h'$. Generally, the start state is independent of the transition matrix $\theta$ in HMM, i.e., $q(h'|\theta) = q(h')$. Moreover, the start state $h'$ is independent of the delimiter hidden state $h^f$ controlled by users. This is because the input-output pairs are sampled independently, where the delimiter states have no effect on the start states of these pairs, i.e., $q(h') = q(h'|h^f)$. Detailed discussions can be found in the Appendix D.

### 3.3 ASSUMPTION

We formalize some priors and understanding of ICL by introducing the following assumptions, which form the basis for our main theorem. For instance, without loss of generality, we have no priors in the text generation process to determine which token is forbidden, leading to Assumption 1.

**Assumption 1 (Token Priors)** *All tokens are available, i.e., $\forall o \in \mathcal{O}, \forall \theta \in \Theta, \exists h \in \mathcal{H}, p(o|h, \theta) > \delta_{1,1} > 0$. Delimiter tokens can be sampled from delimiter states, i.e., $p(o^f|h^f) > \delta_{1,2} > 0$.*

The transition may occur between any state. Thus, similar to the token priors, no priors are known for determining which transition is forbidden for text generation, as described in Assumption 2.

**Assumption 2 (Hidden State Priors)** *No priors of forbidden transitions between states are known: $\forall h, \tilde{h} \in \mathcal{H}, \theta \in \Theta, p(h|\tilde{h}, \theta) > \delta_{2,1} > 0$. Delimiter states form a subset of $\mathcal{H}$: $p(h^f) < \delta_{2,2} < 1$.*

Generally, intentions have their support over the entire intention family $\Theta$. Meanwhile, the intuition of applying ICL to a certain task is that $\Theta$ can cover the ground-truth intention of the task[2].

---

[2]Exploring the scenario $\theta_g \notin \Theta$ is challenging and inherent to the training process of LLMs. In our future work, we will investigate whether LLMs can infer with unseen intentions.

**Assumption 3 (Intention Priors)** *The intention has support over $\Theta$: $\forall \theta \in \Theta, p(\theta) > 0$. Similar intentions produce a bounded discrepancy between token distributions: $\forall \epsilon > 0, \exists\, \delta_3, \forall \theta, \tilde{\theta} \in \Theta$, if $KL\left(q\left(x, y|\tilde{\theta}\right) || q\left(x, y|\theta\right)\right) < \epsilon$, we have $|p(o_i|o_{1:i-1}, \theta) - p(o_i|o_{1:i-1}, \tilde{\theta})| < \delta_3$. The intention family covers the ground-truth intention, i.e., $\theta_g \in \Theta$.*

According to the objective function for training LLMs, the error of next-token predicting is minimized. Thus, it is intuitive to assume that the prediction errors of LLMs are bounded on general tasks and that LLMs can predict the delimiters used in the demonstration format. Formal descriptions are given in the following Assumption 4.

**Assumption 4 (LLMs Priors)** *LLMs can predict the next token with a bounded error: $\forall \theta \in \Theta$, we have $p_M(\theta) > 0$ and $|p(o_i|o_{1:i-1}, \theta) - p_M(o_i|o_{1:i-1}, \theta)| < \delta_{4,1}$. LLMs can predict tokens by predicting delimiter states: $\forall h^f \in \mathcal{H}^f, \forall h \in \mathcal{H}, \forall \theta, p_M(h^f|h) \in [\delta_{4,2}, \delta_{4,3}]$ and $p_M(o|h, \theta) < \delta_{4,4}$.*

Assumption 4 only considers cases where the model infers a relatively correct intention, and if the model infers a wrong intention, i.e., we expect this probability to be small and bounded. Thus, this assumption is loose. Besides the LLMs priors, we further assume that the LLMs can produce similar start tokens when conditioning on similar intentions, i.e., prediction smoothness, as shown in the following Assumption 5.

**Assumption 5 (Prediction Smoothness)** *Similar intentions lead to a bounded discrepancy: $\forall \epsilon > 0, \exists\, \delta_5, \forall \theta, \tilde{\theta} \in \Theta, KL\left(q\left(x, y|\tilde{\theta}\right) || q\left(x, y|\theta\right)\right) < \epsilon, |p_M(o'_i|o^f_{i-1}, \theta) - p_M(o'_i|o^f_{i-1}, \tilde{\theta})| < \delta_5$.*

Note that the above assumptions are weak and mainly introduced to formalize the priors and prediction errors of ICL quantitatively.

## 4 EXPLAINING IN-CONTEXT LEARNING WITH INTENTION MODEL

In this section, we will analyze the prediction errors of an in-context predictor and show its connection to concepts in Sec. 3.

### 4.1 TARGET OVERVIEW

LLMs are mainly trained with a simple objective function that encourages models to predict the next token over a large corpus. As LLMs scale up, ICL emerges as a surprising capability. ICL enables LLMs to learn to do a new task by simply conditioning on a few input-output demonstrations of the task. This exciting and newly discovered capability introduces several intriguing questions: *When does ICL emerge? Why does ICL emerge despite the absence of explicit training for such behavior? How does ICL remain effective when demonstrations differ from LLMs' training distributions?*

Let us start with a general understanding of ICL. It is known that ICL performance mainly relies on two factors: i) LLMs's performance and ii) demonstration quality. This is intuitive because the discrepancy between the predicted and ground-truth distribution stems from these two factors. Namely, the predicted distribution obtained by LLMs through ICL can be written as $q_M(y|S^n(\theta_d), x_t)$, where the corresponding ground-truth distribution is $q(y|S^n(\theta_g), x_t)$. Thus, it would benefit to fulfill these questions about when, why, and how ICL emerges if we can model the discrepancy between these two distributions and identify conditions when the discrepancy is bounded.

With the prediction expansion $p_M(y|S^n(\theta_d), x_t) = \int_\Theta p_M\left(y|S^n\left(\theta_d\right), x_t, \theta\right) p_M\left(\theta|S^n\left(\theta_d\right), x_t\right) d\theta$, we can analyze the discrepancy. The first term $p_M\left(y|S^n\left(\theta_d\right), x_t, \theta\right)$ characterizes LLMs' ability to make predictions with an intention. Meanwhile, the second term $p_M\left(\theta|S^n\left(\theta_d\right), x_t\right)$ shows the ability to infer an intention from demonstrations. Ideally, we expect LLMs to i) produce the ground-truth output: $p_M\left(y|S^n\left(\theta_d\right), x_t, \theta_g\right) \rightarrow p(y|x_t, \theta_g)$, and ii) infer ground-truth intention: $\forall \theta \neq \theta_g, p_M\left(\theta|S^n\left(\theta_d\right), x_t\right) \rightarrow 0$. In this context, we have: $p_M\left(y|S^n\left(\theta_d\right), x_t\right) = p(y|x_t, \theta_g)$.

## 4.2 No Free Lunch Theorem for ICL

**Intention error.** The probability of predicting an unrelated intention, i.e., $\theta \notin \Theta_\epsilon$, can be written as,

$$p_M(\theta|S^n(\theta_d), x_t) = p_M(\theta) \frac{p(S^n(\theta_d), x_t)}{p_M(S^n(\theta_d), x_t)} \frac{p(\theta_g|S^n(\theta_d), x_t)}{p(\theta_g)} \exp\left(n \cdot r(n, \theta)\right), \theta \notin \Theta_\epsilon, \quad (5)$$

where $p_M(\theta)$ is the intention prior of LLMs, $\frac{p(S^n(\theta_d), x_t)}{p_M(S^n(\theta_d), x_t)}$ denotes the ratio of priors of prompts sampled from two different distributions, $\frac{p(\theta_g|S^n(\theta_d), x_t)}{p(\theta_g)}$ represents the ratio of the probability of sampling the ground-truth intention $\theta_g$ with and without prompts from the demonstration intention, $r(n, \theta) \triangleq \frac{1}{n}(\log \frac{p_M(S^n(\theta_d), x_t|\theta)}{p(S^n(\theta_d), x_t|\theta_g)})$ is a function of unrelated intention, demonstrations, and test input. More details are in Appendix B, where we show that $r(n, \theta)$ has an upper bound as follows,

$$r(n, \theta) < \underbrace{\delta}_{\substack{\text{demonstration} \\ \text{shift}}} - \underbrace{a_1 \text{KL}(q_M(x, y|\theta_d)||q_M(x, y|\theta))}_{\substack{\text{distribution discrepancy} \\ \text{induced by unrelated intention } \theta}} + \underbrace{\log a_2}_{\text{constant}} + \underbrace{\frac{1}{n}\log a_3}_{\substack{\text{inverse linear} \\ \text{convergence w.r.t. } n}} + \underbrace{\Delta(n)}_{\substack{\text{estimation error} \\ \text{converge w.r.t. } n}}, \theta \notin \Theta_\epsilon,$$

$$(6)$$

where $\delta = \text{KL}(q(x, y|\theta_d)||q(x, y|\theta_g))$ represents the deviation between demonstration and ground-truth intention, $a_1 = \frac{\delta_{2,1}^{2k-1}\delta_{1,1}^{2k}}{\delta_{4,4}^{2k}} > 0$, $a_2 = \frac{\delta_{4,3}\delta_{2,2}\delta_{4,4}^{2k}}{\delta_{4,2}\delta_{2,1}\delta_{2,1}^{2k-1}\delta_{1,1}^{2k}} > 0$, $a_3 = \frac{\delta_{4,4}^k}{\delta_{1,2}\delta_{2,1}^{k-1}\delta_{1,1}^k} > 0$ are constant and $\Delta(n)$ approaches 0 as the number of demonstrations $n$ increases. Let $\log \frac{\delta_{4,3}\delta_{2,2}\delta_{4,4}^{2k}}{\delta_{4,2}\delta_{2,1}\delta_{2,1}^{2k-1}\delta_{1,1}^{2k}} + \delta - \frac{\delta_{2,1}^{2k-1}\delta_{1,1}^{2k}}{\delta_{4,4}^{2k}}\text{KL}(q_M(O|\theta_d)||q_M(O|\theta)) \triangleq F(\theta)$ we have $F(\theta_d) > 0$. Due to the discrepancy between intention $\theta_d$ and unrelated intentions $\theta$ and the continuity of $F(\theta_d)$, there exists an $\epsilon$ such that $\forall \theta$ s.t. $\text{KL}(q_M(x, y|\theta_g)||q_M(x, y|\theta)) > \epsilon$, $F(\theta) < \frac{m}{2}, m < 0$, where m is the upper bound of the function $F(\theta_d)$ on the boundary. Thus, unrelated intentions lead to $exp(n \cdot r(n, \theta)) < C \cdot exp(n \cdot \frac{m}{2})$, namely, $\exp(n \cdot r(n, \theta)) \to 0$ given a sufficient number of demonstrations $n$. Accordingly, LLMs will filter out these unrelated intentions, i.e., $p_M(\theta|S^n(\theta_d), x_t) \to 0$.

**Output error.** Based on the inferred intention $\theta_g$, we show in the Appendix C.1 that LLMs can predict a surrogate outputs $p(y|S^n(\theta_d), x_t, \theta_g)$ with bounded prediction error $\eta_e$,

$$\eta_e \triangleq |p_M(y|S^n(\theta_d), x_t, \theta_g) - p(y|S^n(\theta_d), x_t, \theta_g)| < k_t\delta_{4,1} + \mathcal{O}(\delta_{4,1}), \quad (7)$$

where $k_t$ is the number of tokens in the output $y_t(\theta_g)$ and $\mathcal{O}(\delta_{4,1})$ is the higher-order error of $\delta_{4,1}$.

In ICL, the prediction of an LLm is $p(y|S^n(\theta_d), x_t, \theta_g)$ rather than $p_M(y|S^n(\theta_d), x_t, \theta_g)$. Thus, there are still prediction errors when $p_M(y|S^n(\theta_d), x_t, \theta_g)$ approaches to the ground-truth. Detailed discussions and derivations can be found in Appendix C.2. Specifically, the first prediction noise $\eta_{n_1}$ induced by the second term of expansion in Eq. 78 is bounded,

$$\eta_{n_1} \triangleq \int_{\Theta \backslash \Theta_\epsilon} p_M(y|S, x_t, \theta)p_M(\theta)p_M(S, x_t|\theta)d\theta < C \cdot \exp\left(n \cdot \frac{m}{2}\right), \quad (8)$$

where $m < 0$ (see Appendix B) implies that increasing the number of demonstrations $n$ leads to rapidly decreasing prediction noise $\eta_{n_1}$. This aligns with our understanding of the ICL capability.

As shown in Appendix C.3, the last term in the expansion causes bounded prediction noise $\eta_{n_2}$,

$$\eta_{n_2} < \left(\underbrace{(2k_t - 1 + 2kn - n)2\delta_{4,1}}_{\text{error in predicting next token}} + \underbrace{n\delta_5}_{\substack{\text{prediction} \\ \text{smoothness}}} + \underbrace{(2k_t - 1 + 2kn - n)\delta_3}_{\text{distribution smoothness}} + \underbrace{\mathcal{O}(1)}_{\substack{\text{higher order} \\ \text{error}}}\right)\underbrace{p_M(\Theta_\epsilon)}_{\substack{\text{demonstration} \\ \text{shift}}}, \quad (9)$$

where $2k_t$ is the number of tokens of the test input-output pair, $2kn$ the number of tokens of demonstrations, $n$ the number of pairs in demonstrations, $\delta_{4,1}$ the prediction error of the next token depending on LLMs, $\delta_5$ the prediction smoothness of LLMs, $\delta_3$ a constant about the intention priors, $\mathcal{O}(1)$ the higher order error, and $p_M(\Theta_\epsilon) = \int_{\Theta_\epsilon} p_M(\theta)d\theta$ captures the demonstration shift. Note that within the intention neighborhood inferred by LLMs, the cumulative effects of the next-token error and the prediction smoothness compound multiplicatively, resulting in a more intricate error term.

To facilitate comprehension of this error, we employ a binomial expansion, reducing higher-order terms to a constant. This approach offers an intuitive grasp of the error term's actual impact.

The first term in Eq. 9 shows that the prediction noise $\eta_{n_2}$ is related to the error in predicting the next token, i.e., $\delta_{4,1}$, increasing with $n$ and $k$. This is intuitive: more demonstrations would magnify the error of LLMs. Prediction smoothness is crucial in the traditional learning paradigm (Bengio et al., 2013). Here, the second term shows that prediction smoothness is also crucial for the ICL capability of LLMs, increasing with $n$. Similarly, the third term shows the error induced by distribution smoothness, which is also increasing with $n$ and $k$. The last component of the error prediction is induced by the process of constructing demonstrations, i.e., demonstration shift. This is intuitive: high-quality demonstrations lead to small $p_M(\Theta_\epsilon)$, further resulting in a small prediction noise $\eta_{n_2}$.

**Theorem 1 (No free lunch theorem for ICL)** *Assume all assumptions in Sec. 3.3 hold. When conditioning on demonstrations, whether the ICL capability of an LLM emerges depends on prediction error $\eta_e$ and prediction noise $\eta_n$, which are determined by three factors: i) LLM's prediction error of the next token, $\delta_{4,1}$, ii) LLM's prediction smoothness, $\delta_5$, and iii) demonstration quality, $\epsilon$ and $n$,*

$$\eta_e < g_e(\delta_{4,1}), \quad \eta_n \triangleq \eta_{n_1} + \eta_{n_2} < g_1(n) + g_2(\delta_{4,1}, \delta_5, n, \epsilon),$$

*where $g_e(\delta_{4,1}) \triangleq k_t\delta_{4,1} + \mathcal{O}(1)$ with $k_t$ the number of tokens in the output and a higher order error $\mathcal{O}(1)$, $g_1(n) \triangleq C \cdot \exp\left(n \cdot \frac{m}{2}\right)$ with $m < 0$, and $g_2(\delta_{4,1}, \delta_5, n, \epsilon) \triangleq ((2k_t - 1 + 2kn - n)\delta_{4,1} + n\delta_5 + (2kn - n + 2k_t - 1)\delta_3 + \mathcal{O}(1)) p_M(\Theta_\epsilon)$.*

The proposed no-free-lunch Theorem 1 for ICL indicates that the prediction mismatch between the ground-truth and in-context predictor outputs is determined by LLMs' performance and the quality of demonstrations. Note that, we use the *no-free-lunch* here not to emphasize a universal algorithm but rather a specific condition for the existence of ICL capability. Our theoretical results provide a general theoretical basis that general capability of predicting the next token and the demonstration shift from the task intention plays a crucial role in ICL. LLMs with weak capability in predicting the next token and non-qualified demonstrations fail to exhibit ICL capability. This is consistent with the understanding in the literature, i.e., ICL capability increases with LLMs' scale (Brown et al., 2020), and demonstrations play a crucial role in ICL (Dong et al., 2023).

### 4.3 EXPLAINING ICL WITH INTENTION MODEL

It shows that ICL can achieve good performance with randomly assigned labels (Min et al., 2022), implying that ground-truth input-output mappings are not required. In contrast, a novel perspective (Kossen et al., 2024) is then proposed: ICL predictions depend on the input-output mapping of demonstrations. Moreover, it is shown that larger LLMs can override semantic priors, i.e., following the flipped labels in demonstrations (Wei et al., 2023). In the flipped label scenario, demonstrations are constructed to describe a binary classification task in which all labels of input samples are flipped. These correct but differing views make understanding and explaining ICL difficult. Fortunately, arming with the intention model, we will find that these outstanding observations are correct and predictable under specific conditions.

Our intention model provides a novel view of explaining the input-output mapping in ICL. In these scenarios, the original outputs can be wrapped in the way of multiplying a transition matrix by the original transition matrix, e.g., applying an external mapping to the original outputs,

$$y_t(\mathcal{T}\theta_g) = \arg\max_y p(y|x_t, \mathcal{T}\theta_g), \text{with } y(\mathcal{T}\theta_d) = \arg\max_y p(y|x, \mathcal{T}\theta_d), \tag{10}$$

where $\mathcal{T}$ is a transition matrix applied to the original transition matrix. Namely, users can apply an external intention operation $\mathcal{T}$ to change the original intention, hoping LLMs can capture the modified intention and behave like humans. For instance, the external matrix could be a function mapping the original output *positive* to a flipped version *negative* or a non-semantic version *foo*. Thus, Eq. 10 formalizes the scenarios of ICL with varying input-output mappings. In this context, we can generalize the flipped label scenario to a scenario, where we can control the complexity of the question by introducing a complex intention operation $\mathcal{T}$. For instance, flipping labels in a binary classification task is equivalent to mapping the output through mode 2 addition, i.e., $y = (y+1) \bmod 2$. Taking a step further, we can instantiate the mapping under multiple-choice problems.

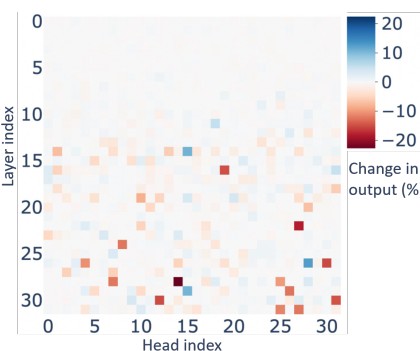 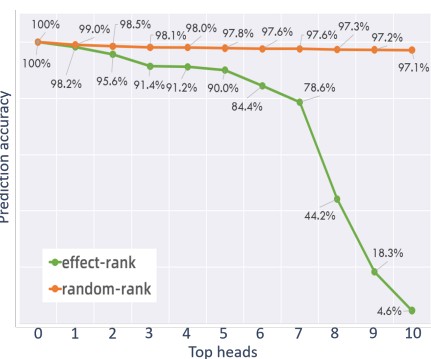

Figure 1: **Left**: induction heads at different layers and positions are visualized in color, denoting the output changes. **Right**: Prediction accuracy varies with the number of knocked-out induction heads, where 10 heads are involved and sorted. Here, the "effect-rank" means the heads are sorted according to the changes in outputs of identified heads, and the "random-rank" is sorted randomly.

Specifically, we can construct demonstrations for a multiple-choice question, where the ground-truth outputs of the original questions are modified through a mode addition operation, i.e., $y = (y + 1)$ mod 5. Similarly, mapping *positive* and *negative* to *bar* and *foo* could be generalized as the scenarios where $(a) \rightarrow apple$, $(b) \rightarrow banana$, $(c) \rightarrow cherry$ for multiple-choice problems.

Intuitively, applying an external operation to the original intention matrix $\theta$ makes the inference more challenging due to the additional estimation for the matrix $\mathcal{T}$. According to the proposed intention model, this is inherently related to the estimation error as shown in Eq. 6 and Eq. 9. Namely, the demonstration shift $\mathrm{KL}(q_M(x, y|\theta_d) \| q_M(x, y|\theta_g)) = \delta < \delta' = \mathrm{KL}(q_M(x, y|\mathcal{T}\theta_d) \| q_M(x, y|\mathcal{T}\theta_g))$ could be larger than before due to the introduced external matrix $\mathcal{T}$. This would lead to larger $r(n, \theta)$. Namely, an extremely complex external matrix would make it challenging to infer the ground-truth intentions and degraded ICL performance. Moreover, introducing the external transition matrix leads to $p_M(\Theta_\epsilon) < p_M(\Theta_{\epsilon'})$, which increases the prediction noise $\eta_{n_2}$ as shown in Eq. 9. Thus, we can conclude that introducing an external to modify the original outputs of demonstrations makes ICL more challenging. This is consistent with our observations, as shown in Table 1.

Eq. 9 shows that an LLM with a slight error of next-token prediction $\delta_{4,1}$ can reduce the prediction noise $\eta_{n_2}$. This is inherently related to overriding the semantic priors. Namely, an LLM with a small error in next-token prediction performs better in overriding semantic priors under flipped label scenarios. This conclusion aligns well with our experimental observations (Table 1) and previous experimental observations (Wei et al., 2023). Increasing the number of demonstrations under the random label scenario decreases performance. This is because a larger number of demonstrations would magnify the impact of demonstration shift and the LLMs' error of next-token prediction, as shown in Eq. 9. This conclusion aligns well with previous experimental observations (Kossen et al., 2024). Analyzing a random mechanism $\mathcal{T}$ is more challenging because each demonstration would be generated with a distinct intention. This is inherently related to the mixed intention scenarios, see Sec. 7. Therefore, we leave theoretical explanations about a random $\mathcal{T}$ as our future work.

## 5 EXPERIMENTS

### 5.1 POSSIBLE MECHANISM OF INTENTION INFERENCE.

The basic idea of our intention model is that LLMs can infer task intentions from the demonstrations and filter out unrelated intention. A natural question is raised: what is the instantiation of intentions in LLMs? According to Olsson et al. (2022), induction heads are shown to be closely related to general ICL in LLMs. Namely, induction heads could be the mechanistic source of general ICL in LLMs Olsson et al. (2022). This motivates us to identify a set of induction heads for a given intention. Thus, we aim to design experiments to verify the impact of these intentions on the generation process under a specific intention.

Intuitively, some specific induction heads mainly correspond to a specific intention. Therefore, we aim to identify a set of induction heads for a given intention and verify the impact of these

intentions on the generation process under a specific intention. For the algorithm to locate induction heads, we draw inspiration from the work Goldowsky-Dill et al. (2023). We construct counterfactual examples to compare the induction heads when the intention of interest is activated and when it is not. Specifically, given a reference sample used for activating a certain intention, we construct a corresponding counterfactual example to deactivate the intention with minimal changes to the reference sample. Subsequently, we replace the induction heads of the reference sample with those of the counterfactual example. Thus, we can record the output changes when replacing each head. Consequently, induction heads that cause drastic output changes are located as the candidate heads related to the current intention. Following previous work, we employ the model LLaMA-2-7B and the dataset SST-2 for the employed models and datasets. More details can be found in Appendix F. Fig. 1 shows the map of identified induction heads. It can be seen that only a small fraction of heads are frequently involved in generation for a specific intention. To verify whether these heads are related to the same intention, we perturb these identified heads of samples, which are different from those used in the intention-locating process. For instance, replacing values of these heads with zeros (Goldowsky-Dill et al., 2023). We also randomly sample an equal number of induction heads for perturbation as a control experiment. The effect of perturbing induction heads on LLMs' performance is shown in Fig. 1. We can see that the heads identified with distinct samples can significantly reduce prediction accuracy when they are perturbed, a phenomenon not observed on randomly selected heads. Thus, these identified induction heads are highly related to the intention.

## 5.2 STUDY OF LEARNING BEHAVIOR

Our theorem shows a no-free-lunch nature of ICL, involving prediction error and noise. A straightforward approach to validate the theorem is to calculate the prediction error and noise. However, it is challenging to calculate or estimate these values, i.e. the *error of next-token prediction $\delta_3$*, *LLM's prediction smoothness*, and *demonstration shift $\epsilon$*. Therefore, it is challenging to provide a quantitive analysis to verify the theorem. Fortunately, some related factors can be controlled implicitly. Thus, to validate our theorem, we design experiments to test the impact of these factors implicitly. For instance, the error of next-token prediction $\delta_3$ could be related to the LLMs' performance under general tasks. In this context, we could conclude that $\delta_3$ of GPT4 is less than that of LLaMa-7B.

Eq. 10 generalizes the flipped-label scenario to the problem of capturing complex mapping relations. We can then validate our theoretical insights by experimenting on flipped label questions. In this regard, we instantiate the mapping using a multiple-choice question sampled from the CSQA dataset (Saha et al., 2018), where the ground-truth outputs of the original questions are changed with a mode addition operation, i.e., i) $y = (y + 1) \mod 5$, and ii) $y = (3y + 1) \mod 5$. According to our theoretical results, larger LLMs are able to follow the instructions shown in the demonstrations. For instance, if the answer to a certain question is "(a): Yes" but labeled as "(b)", larger LLMs can predict "(b): No" as the answer, even though the answer in the option "(b)" does not match the actual answer. As a simple corollary to our theory, learning behaviors under a fliped-label scenario can be modeled by multiplying a transition matrix $\mathcal{T}$ by the original transition matrix $\theta$, as shown in Eq 10. Introducing an external $\mathcal{T}$ to modify original outputs would lead to $\mathrm{KL}(q_M(x, y | \mathcal{T}\theta_d) || q_M(x, y | \mathcal{T}\theta_g)) = \delta' > \delta = \mathrm{KL}(q_M(x, y | \theta_d) || q_M(x, y | \theta_g))$, making the task more challenging. This is becuase large $\epsilon'$ results in 1) larger prediction errors as shown in Eq. (7); and 2) larger prediction noise as shown in Eq. (13). Thus, introducing an external matrix $\mathcal{T}$ will degrade ICL performance, which is consistent with the results shown in Table 1, i.e., changing $y$ to $(y + 1) \mod 5$. Our theoretical result also shows that LLMs with smaller errors of next-token prediction $\delta_3$ perform better in overriding semantic priors under flipped label scenarios. This is because a smaller prediction error $\delta_3$ can reduce the prediction noise, as shown in Eq. 10.

We leverage three LLMs: LLaMa-7B (Touvron et al., 2023), Mistral-7B (Jiang et al., 2023), and GPT-4 (Achiam et al., 2023). The results in Table 1 show that larger LLMs can capture the intention in demonstrations, while the (relatively) small LLM fails to follow the external intentions. This shares the same spirit with the flipped label scenario (Wei et al., 2023). Thus, these experimental results align with our conclusion: an LLM with a minor error of next-token prediction $\delta_3$ performs better in overriding semantic priors under flipped label scenarios.

We observe that GPT-4 achieves exciting performance when changing the label from $y$ to $(y + 1) \mod 5$. However, changing $y$ to $(3y + 1) \mod 5$ drastically degrades its performance, while increasing the number of demonstrations still fails to promote its performance considerably. This

Table 1: Prediction accuracy (%) evaluated with three LLMs on 100 questions from CSQA dataset. Here, $n$ denotes the number of demonstrations defined in Eq. 1.

| $n$ | | $n = 7$ | | $n = 3$ | $n = 14$ |
|---|---|---|---|---|---|
| $y$ | $y$ | $(y + 1) \bmod 5$ | $(3y + 1) \bmod 5$ | $(3y + 1) \bmod 5$ | $(3y + 1) \bmod 5$ |
| LLaMa-7B | 62 | 55 | 36 | 28 | 36 |
| Mistral-7B | 78 | 72 | 60 | 52 | 63 |
| GPT-4 | 100 | 100 | 90 | 88 | 91 |

is inherently related to a new research question about quantitatively measuring the complexity of different external transition matrixes.

## 6 RELATED WORK

Language modeling stands as a pivotal technology in the vast domain of natural language processing (NLP) (Radford et al., 2019; Lewis et al., 2020). Scaling models often exhibit distinct behaviors from their smaller counterparts and display surprising capabilities. Trained with immense text data collections, LLMs (Brown et al., 2020; Touvron et al., 2023; Jiang et al., 2023) have revolutionized various NLP tasks. Unlike traditional transfer learning, LLMs showcase a remarkable ICL capability, a framework that empowers LLMs to grasp tasks through conditioning on some demonstrations (Dong et al., 2023). Namely, LLMs can do a downstream task by simply conditioning on a few input-output demonstrations without model updates. Advanced works (Liu et al., 2022; Wang et al., 2024) show that the label space presented, the overall format of the presentation, and the selection of demonstrations play a crucial role in ICL (Min et al., 2022). Theoretical explorations on the learnability of ICL mainly focused on the analysis of demonstration format in ICL (Xie et al., 2022; Wies et al., 2023). Meanwhile, Lin & Lee (2024) consider a specific regression task and assess the performance of ICL with the focus on context length's impact on ICL and explains two Real-World phenomena. In contrast, our work not only considers the possible errors of the demonstration itself but also considers the model's general forecasting ability. Some empirical understanding (Kossen et al., 2024; Pan et al., 2023) on the mechanism of ICL. We defer detailed discussions in Appendix E.

## 7 LIMITATION.

**Demonstration order** is not covered by our work, while recent works show that the order of demonstrations also plays a crucial role in ICL (Lu et al., 2022; Perez et al., 2021). One possible approach is to introduce the mechanism of ICL like Reddy (2024), which is left as our future work.

**Mixed intention** is not considered in our work. Although generating all input-output pairs with the same intention is intuitive, studying mixed intentions is also a promising direction. Specifically, what would happen when an input-output pair sampled from a different intention is injected into the demonstration is unclear. Thus, we leave it as our future work.

**Explicit intention** is a possible approach to contribute to training LLMs, as our intention model shows that inferring intention plays a crucial role in ICL. Training an LLM is out of the scope of this work, and we hope to verify the point in our future work. Moreover, exploring ICL through the lens of novel tasks is a promising direction not discussed in our work. Learning with in-distribution data often struggles to achieve robust generalization performance on out-of-distribution data, which usually requires further exploration regarding the learnability and generalizability Fang et al. (2022). Thus, we will explore the novel intentions and novel tasks in our future work.

## 8 CONCLUSION

We propose a novel intention model framework to investigate the in-context learning (ICL) capabilities of large language models. In particular, we give a no-free-lunch theorem: whether ICL emerges depends on the prediction error and prediction noise, which are determined by i) LLMs' error of next-token prediction, ii) LLMs' prediction smoothness, and iii) demonstration shift. Note that our theoretical results connect LLMs' error of next-token prediction to the ICL capability. Moreover, our theoretical results bridge the gap between theoretical explanations and empirical observations.

## ETHIC STATEMENT

This paper does not raise any ethical concerns. This study does not involve any human subjects, practices to data set releases, potentially harmful insights, methodologies and applications, potential conflicts of interest and sponsorship, discrimination/bias/fairness concerns, privacy and security issues, legal compliance, and research integrity issues.

This paper is intended to theoretically explain the in-context learning capability of LLMs, hoping to connect the theoretical explanations and empirical understandings. The study does not introduce risks or adverse consequences that could result from its implementation.

## REPRODUCIBILITY STATEMENT

To ensure the reproducibility of our theoretical and empirical results, we have provided detailed descriptions of our derivations and implementation details. We also summarize our efforts below to facilitate reproducible results:

**Theoretical results.** Statements of the theoretical results can be found in Appendix A, B, and C.

**Datasets.** We use publicly available datasets described in detail in Appendix F.

**Open Source.** Code will be available once the paper is accepted.

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

# A    SOME PROPOSITIONS

Firstly, we propose some propositions, which are corollaries of the assumptions in Sec. 3.3.

**Proposition 1.** $\forall i \in \{1, \cdots, k\}$, we have $p(h_i^f | O_1, o_1^f, \cdots, O_i, \theta) > \delta_{2,1}$. Meanwhile, $\forall i \in \{1, \cdots, k\}, \forall \theta \notin \Theta_\epsilon$, we have $p_M(h_i^f | O_1, o_1^f, \cdots, O_i, \theta) < \delta_{4,3}$, where $O_i \triangleq [x_i, y_i] = [o_{i,1}, o_{i,2}, \cdots, o_{i,2k}]$ stands for the $i$-th input-output pair with $2k$ tokens.

Proof: By Assumption 2, we have:

$$p(h_i^f | O_1, o_1^f, \cdots, O_i, \theta) = \sum_h p(h_i^f | h, \theta) p(h | O_1, o_1^f, \cdots, O_i, \theta) \tag{11}$$

$$= \sum_h p(h_i^f | h) p(h | O_1, o_1^f, \cdots, O_i, \theta) \tag{12}$$

$$> \sum_h \delta_{2,1} p(h | O_1, o_1^f, \cdots, O_i, \theta) = \delta_{2,1}. \tag{13}$$

Similarly, by Assumption 4, $\forall \theta \notin \Theta_\epsilon$, we have:

$$p_M(h_i^f | O_1, o_1^f, \cdots, O_i, \theta) = \sum_h p_M(h_i^f | h, \theta) p_M(h | O_1, o_1^f, \cdots, O_i, \theta) \tag{14}$$

$$= \sum_h p_M(h_i^f | h) p_M(h | O_1, o_1^f, \cdots, O_i, \theta) \tag{15}$$

$$< \sum_h \delta_{4,3} p_M(h | O_1, o_1^f, \cdots, O_i, \theta) = \delta_{4,3}. \tag{16}$$

**Proposition 2.** $\forall O_i = [x_i, y_i] = \{\mathcal{D}_{x_i}^k, \mathcal{D}_{y_i}^k\}, \forall \theta \in \Theta$, we have $p(O_i | h_i', \theta) > \delta_{2,1}^{2k-1} \delta_{1,1}^{2k}$.

Proof: From Assumption 1 and Assumption 2, we have:

$$p(O_i | h_i', \theta) = \prod_{j=2}^{2k} \sum_{h \in \mathcal{H}} p(o_{i,j} | h, \theta) p(h | h_{i,j-1}, \theta) \, p(o_{i,1} | h_i', \theta) > \delta_{2,1}^{2k-1} \delta_{1,1}^{2k}. \tag{17}$$

Similarly, we have $p(x_i | h_i', \theta) > \delta_{2,1}^{k-1} \delta_{1,1}^k$

**Proposition 3.** $\forall O_i = [x_i, y_i] = \{\mathcal{D}_{x_i}^k, \mathcal{D}_{y_i}^k\}, \forall \theta \in \Theta$, we have $p_M(O_i | h_i', \theta) < \delta_{4,4}^{2k}$.

Proof: From Assumption 3, we have:

$$p_M(O_i | h_i', \theta) = \prod_{j=2}^{2k} \sum_{h \in \mathcal{H}} p_M(o_{i,j} | h, \theta) p_M(h | h_{i,j-1}, \theta) \, p_M(o_{i,1} | h_i', \theta) < \delta_{4,4}^{2k}, \tag{18}$$

where $h_i' = h_{i,1}$ represents the start hidden state. Similarly, we have $p(x_i | h_i', \theta) < \delta_{4,4}^k$

# B    INTENTION ERROR PROBABILITY OF LLMS

In this section, we will theoretically analyze the probability of the LLMs predicting the unrelated intention, i.e., $\theta \notin \Theta_\epsilon$. To emphasize the number of examples, we use $S^n$ to represent $S^n(\theta_d)$.

For $\theta \notin \Theta_\epsilon$, we consider that $p_M(S^n, x_t, \theta) > 0$, otherwise $p_M(S^n, x_t, \theta) = 0$, LLMs will not launch $\theta$. To facilitate our analysis, we define $\frac{p_M(S^n, x_t | \theta)}{p(S^n, x_t | \theta_g)} = e^{n \cdot \frac{1}{n} \log \frac{p_M(S^n, x_t | \theta)}{p(S^n, x_t | \theta_g)}} = e^{n \cdot r(n, \theta)}$. Let

$O_i^{ex} = [o_{i-1}^f \ x_i \ y_i], O_1^{ex} = [x_1 \ y_1], O_{1:0}^{ex} \triangleq \emptyset$. Then, we have:

$$p_M(S^n, x_t | \theta) = p_M(x_t | S^n, \theta) p_M(S^n | \theta) \tag{19}$$

$$= p_M(x_t | S^n, \theta) \sum_{h_n^f \in \mathcal{H}^f} p_M(o_n^f | h_n^f) p_M(h_n^f | O_{1:n}^{ex}, \theta)$$

$$\prod_{i=1}^{n} \sum_{h_{i-1}^f \in \mathcal{H}^f} p_M(O_i | h_{i-1}^f, \theta) p_M(h_{i-1}^f | O_{1:i-1}^{ex}, \theta) \tag{20}$$

$$\leq \sum_{h_n^f \in \mathcal{H}^f} \delta_{4,4}^k p_M(h_n^f | O_{1:n}^{ex}, \theta) \prod_{i=1}^{n} \sum_{h_{i-1}^f \in \mathcal{H}^f} p_M(O_i | h_{i-1}^f, \theta) p_M(h_{i-1}^f | O_{1:i-1}^{ex}, \theta) \tag{21}$$

$$= \prod_{i=1}^{n} \sum_{h_{i-1}^f \in \mathcal{H}^f} p_M(O_i | h_{i-1}^f, \theta) p_M(h_{i-1}^f | O_{1:i-1}^{ex}, \theta). \tag{22}$$

Using Assumption 1, we can expand $p(S^n, x_t | \theta)$ similarly:

$$p(S^n, x_t | \theta) = \sum_{h_t' \in \mathcal{H}} p(x_t | h_t', \theta) p(h_t' | S^n, \theta)$$

$$\sum_{h_n^f \in \mathcal{H}^f} p(o_n^f | h_n^f) p(h_n^f | O_{1:n}^{ex}, \theta) \prod_{i=1}^{n} \sum_{h_{i-1}^f \in \mathcal{H}^f} p(O_i | h_{i-1}^f, \theta) p(h_{i-1}^f | O_{1:i-1}^{ex}, \theta) \tag{23}$$

$$> \sum_{h_t' \in \mathcal{H}} p(x_t | h_t', \theta) p(h_t' | S^n, \theta)$$

$$\sum_{h_n^f \in \mathcal{H}^f} \delta_{1,2} \, p(h_n^f | O_{1:n}^{ex}, \theta) \prod_{i=1}^{n} \sum_{h_{i-1}^f \in \mathcal{H}^f} p(O_i | h_{i-1}^f, \theta) p(h_{i-1}^f | O_{1:i-1}^{ex}, \theta) \tag{24}$$

$$= \delta_{1,2} \sum_{h_t' \in \mathcal{H}} p(x_t | h_t', \theta) p(h_t' | S^n, \theta) \prod_{i=1}^{n} \sum_{h_{i-1}^f \in \mathcal{H}^f} p(O_i | h_{i-1}^f, \theta) p(h_{i-1}^f | O_{1:i-1}^{ex}, \theta). \tag{25}$$

Then we can write $r(n, \theta)$ as follows:

$$r(n, \theta) = \frac{1}{n} \log \frac{p_M(S^n, x_t | \theta)}{p(S^n, x_t | \theta_g)} \tag{26}$$

$$< \frac{1}{n} \left( \log \frac{\delta_{4,4}^k}{\delta_{1,2} \sum_{h_t' \in \mathcal{H}} p(x_t | h_t', \theta_g) p(h_t' | S^n, \theta_g)} \right.$$

$$\left. + \sum_{i=1}^{n} \log \frac{\sum_{h_{i-1}^f \in \mathcal{H}^f} p_M(O_i | h_{i-1}^f, \theta) p_M(h_{i-1}^f | O_{1:i-1}^{ex}, \theta)}{\sum_{h_{i-1}^f \in \mathcal{H}^f} p(O_i | h_{i-1}^f, \theta_g) p(h_{i-1}^f | O_{1:i-1}^{ex}, \theta_g)} \right) \tag{27}$$

$$\leq \frac{1}{n} \left( \log(\frac{\delta_{4,4}^k}{\delta_{1,2} \delta_{2,1}^{k-1} \delta_{1,1}^k}) + n \log \frac{\delta_{4,3}}{\delta_{2,1}} + \sum_{i=1}^{n} \log \frac{\sum_{h_{i-1}^f \in \mathcal{H}^f} p_M(O_i | h_{i-1}^f, \theta)}{\sum_{h_{i-1}^f \in \mathcal{H}^f} p(O_i | h_{i-1}^f, \theta_g)} \right). \tag{28}$$

Furthermore, we calculate $\sum_{h_{i-1}^f \in \mathcal{H}^f} p_M(O_i | h_{i-1}^f, \theta)$:

$$\sum_{h_{i-1}^f \in \mathcal{H}^f} p_M(O_i | h_{i-1}^f, \theta) = \sum_{h_{i-1}^f \in \mathcal{H}^f} \sum_{h_i'} p_M(O_i | h_i', \theta) p_M(h_i' | h_{i-1}^f, \theta) \tag{29}$$

$$= \sum_{h_i'} p_M(O_i | h_i', \theta) p_M(h_i' | \theta) \sum_{h_{i-1}^f \in \mathcal{H}^f} \frac{p_M(h_i' | h_{i-1}^f, \theta)}{p_M(h_i' | \theta)} \tag{30}$$

$$= \sum_{h_i'} p_M(O_i | h_i', \theta) p_M(h_i' | \theta) \sum_{h_{i-1}^f \in \mathcal{H}^f} \frac{p_M(h_{i-1}^f | h_i', \theta)}{p_M(h_{i-1}^f | \theta)} \tag{31}$$

$$= p_M(O_i | \theta) \sum_{h_{i-1}^f \in \mathcal{H}^f} \frac{p_M(h_{i-1}^f | h_i', \theta)}{p_M(h_{i-1}^f | \theta)}. \tag{32}$$

The hidden state of the delimiter token is independent of intention, by Assumption 4,

$$\frac{p_M(h_{i-1}^f | h_i', \theta)}{p_M(h_{i-1}^f | \theta)} < \frac{p_M(h_{i-1}^f | h_i', \theta)}{\delta_{4,2}}, \tag{33}$$

And then we can get:

$$\sum_{h_{i-1}^f \in \mathcal{H}^f} p_M(O_i | h_{i-1}^f, \theta) < \sum_{h_{i-1}^f \in \mathcal{H}^f} p_M(O_i | \theta) \frac{p_M(h_{i-1}^f | h_i', \theta)}{\delta_{4,2}} = \frac{1}{\delta_{4,2}} p_M(O_i | \theta). \tag{34}$$

Similarly, with Assumption 2,

$$\frac{p(h_{i-1}^f | h_i', \theta)}{p(h_{i-1}^f | \theta)} > \frac{p(h_{i-1}^f | h_i', \theta)}{\delta_{2,2}}, \tag{35}$$

We can launch:

$$\sum_{h_{i-1}^f \in \mathcal{H}^f} p(O_i | h_{i-1}^f, \theta) > \sum_{h_{i-1}^f \in \mathcal{H}^f} p(O_i | \theta) \frac{p(h_{i-1}^f | h_i', \theta)}{\delta_{2,2}} = \frac{1}{\delta_{2,2}} p(O_i | \theta). \tag{36}$$

Therefore, we have:

$$r(n,\theta) < \frac{1}{n}\log\left(\frac{\delta_{4,4}^k}{\delta_{1,2}\delta_{2,1}^{k-1}\delta_{1,1}^k}\right) + \log\frac{\delta_{4,3}}{\delta_{2,1}} + \log\frac{\delta_{2,2}}{\delta_{4,2}} + \frac{1}{n}\sum_{i=1}^n \log\frac{p_M(O_i|\theta)}{p(O_i|\theta_g)} \tag{37}$$

$$=\frac{1}{n}\log\left(\frac{\delta_{4,4}^k}{\delta_{1,2}\delta_{2,1}^{k-1}\delta_{1,1}^k}\right) + \log\frac{\delta_{4,3}\delta_{2,2}}{\delta_{4,2}\delta_{2,1}} + \frac{1}{n}\sum_{i=1}^n\left(\log\frac{p_M(O_i|\theta_d)}{p(O_i|\theta_g)} - \log\frac{p_M(O_i|\theta_d)}{p_M(O_i|\theta)}\right) \tag{38}$$

$$=\frac{1}{n}\log\left(\frac{\delta_{4,4}^k}{\delta_{1,2}\delta_{2,1}^{k-1}\delta_{1,1}^k}\right) + \log\frac{\delta_{4,3}\delta_{2,2}}{\delta_{4,2}\delta_{2,1}} + \frac{1}{n}\sum_{i=1}^n\left(\log\frac{p_M(O_i|\theta_d)}{p(O_i|\theta_d)} + \log\frac{p(O_i|\theta_d)}{p(O_i|\theta_g)} - \log\frac{p_M(O_i|\theta_d)}{p_M(O_i|\theta)}\right) \tag{39}$$

$$<\frac{1}{n}\log\left(\frac{\delta_{4,4}^k}{\delta_{1,2}\delta_{2,1}^{k-1}\delta_{1,1}^k}\right) + \log\frac{\delta_{4,3}\delta_{2,2}\delta_{4,4}^{2k}}{\delta_{4,2}\delta_{2,1}\delta_{2,1}^{2k-1}\delta_{1,1}^{2k}} + \frac{1}{n}\sum_{i=1}^n\left(\log\frac{p(O_i|\theta_d)}{p(O_i|\theta_g)} - \log\frac{p_M(O_i|\theta_d)}{p_M(O_i|\theta)}\right) \tag{40}$$

$$=\frac{1}{n}\log\left(\frac{\delta_{4,4}^k}{\delta_{1,2}\delta_{2,1}^{k-1}\delta_{1,1}^k}\right) + \log\frac{\delta_{4,3}\delta_{2,2}\delta_{4,4}^{2k}}{\delta_{4,2}\delta_{2,1}\delta_{2,1}^{2k-1}\delta_{1,1}^{2k}} + \frac{1}{n}\sum_{i=1}^n\left(\log\frac{p(O_i|\theta_d)}{p(O_i|\theta_g)} - \log\frac{p_M(O_i|\theta_d)}{p_M(O_i|\theta)}\right)$$
$$+ \mathbb{E}_O\log\frac{p(O|\theta_d)}{p(O|\theta_g)} - \mathbb{E}_O\log\frac{p(O|\theta_d)}{p(O|\theta_g)} \tag{41}$$

$$\leq\frac{1}{n}\log\left(\frac{\delta_{4,4}^k}{\delta_{1,2}\delta_{2,1}^{k-1}\delta_{1,1}^k}\right) + \log\frac{\delta_{4,3}\delta_{2,2}\delta_{4,4}^{2k}}{\delta_{4,2}\delta_{2,1}\delta_{2,1}^{2k-1}\delta_{1,1}^{2k}} + \mathbb{E}_O\log\frac{p(O|\theta_d)}{p(O|\theta_g)} - \frac{1}{n}\sum_{i=1}^n\log\frac{p_M(O_i|\theta_d)}{p_M(O_i|\theta)}$$
$$+ |\frac{1}{n}\sum_{i=1}^n\log\frac{p(O_i|\theta_d)}{p(O_i|\theta_g)} - \mathbb{E}_O\log\frac{p(O|\theta_d)}{p(O|\theta_g)}| \tag{42}$$

$$<\frac{1}{n}\log\left(\frac{\delta_{4,4}^k}{\delta_{1,2}\delta_{2,1}^{k-1}\delta_{1,1}^k}\right) + \log\frac{\delta_{4,3}\delta_{2,2}\delta_{4,4}^{2k}}{\delta_{4,2}\delta_{2,1}\delta_{2,1}^{2k-1}\delta_{1,1}^{2k}} + \delta - \frac{1}{n}\sum_{i=1}^n\log\frac{p_M(O_i|\theta_d)}{p_M(O_i|\theta)} + \Delta(n) \tag{43}$$

$$=\frac{1}{n}\log\left(\frac{\delta_{4,4}^k}{\delta_{1,2}\delta_{2,1}^{k-1}\delta_{1,1}^k}\right) + \log\frac{\delta_{4,3}\delta_{2,2}\delta_{4,4}^{2k}}{\delta_{4,2}\delta_{2,1}\delta_{2,1}^{2k-1}\delta_{1,1}^{2k}} + \delta - \frac{1}{n}\sum_{i=1}^n\log\frac{p_M(O_i|\theta_d)}{p_M(O_i|\theta)} + \Delta(n)$$
$$+ \mathbb{E}_O\log\frac{p_M(O|\theta_d)}{p_M(O|\theta)} - \mathbb{E}_O\log\frac{p_M(O|\theta_d)}{p_M(O|\theta)} \tag{44}$$

$$\leq\frac{1}{n}\log\left(\frac{\delta_{4,4}^k}{\delta_{1,2}\delta_{2,1}^{k-1}\delta_{1,1}^k}\right) + \log\frac{\delta_{4,3}\delta_{2,2}\delta_{4,4}^{2k}}{\delta_{4,2}\delta_{2,1}\delta_{2,1}^{2k-1}\delta_{1,1}^{2k}} + \delta - \mathbb{E}_O\log\frac{p_M(O|\theta_d)}{p_M(O|\theta)} + \Delta(n)$$
$$+ |\mathbb{E}_O\log\frac{p_M(O|\theta_d)}{p_M(O|\theta)} - \frac{1}{n}\sum_{i=1}^n\log\frac{p_M(O_i|\theta_d)}{p_M(O_i|\theta)}| \tag{45}$$

$$<\frac{1}{n}\log\left(\frac{\delta_{4,4}^k}{\delta_{1,2}\delta_{2,1}^{k-1}\delta_{1,1}^k}\right) + \log\frac{\delta_{4,3}\delta_{2,2}\delta_{4,4}^{2k}}{\delta_{4,2}\delta_{2,1}\delta_{2,1}^{2k-1}\delta_{1,1}^{2k}} + \delta - \frac{\delta_{2,1}^{2k-1}\delta_{1,1}^{2k}}{\delta_{4,4}^{2k}}\mathbb{E}_{O_M}\log\frac{p_M(O|\theta_d)}{p_M(O|\theta)} + \Delta(n) \tag{46}$$

$$=\frac{1}{n}\log\left(\frac{\delta_{4,4}^k}{\delta_{1,2}\delta_{2,1}^{k-1}\delta_{1,1}^k}\right) + \log\frac{\delta_{4,3}\delta_{2,2}\delta_{4,4}^{2k}}{\delta_{4,2}\delta_{2,1}\delta_{2,1}^{2k-1}\delta_{1,1}^{2k}} + \delta - \frac{\delta_{2,1}^{2k-1}\delta_{1,1}^{2k}}{\delta_{4,4}^{2k}}\mathrm{KL}(q_M(O|\theta_d)||q_M(O|\theta)) + \Delta(n), \tag{47}$$

where $\frac{1}{n}\log\left(\frac{\delta_{4,4}^k}{\delta_{1,2}\delta_{2,1}^{k-1}\delta_{1,1}^k}\right)$ decreases as the number of demonstrations $n$ increases, $\log\frac{\delta_{4,3}\delta_{2,2}\delta_{4,4}^k}{\delta_{4,2}\delta_{2,1}\delta_{2,1}^{k-1}\delta_{1,1}^k}$ is a constant related to the system, $\mathrm{KL}(q_M(O|\theta_d)||q_M(O|\theta))$ measures the distribution discrepancy between the LLMs' prediction of demonstration intention $\theta_d$ and the unrelated intention $\theta \notin \Theta_\epsilon$, $\delta$ stands for the difference between ground-truth intention $\theta_g$ and demonstration in tension $\theta_d$, and $\Delta(n)$ represents the estimation error of the sampling decreasing as $n$ increases.

According to the upper bound, $r(n, \theta)$ decreases as the number of demonstrations $n$ increases. In this context, we have:

$$\lim_{n \to \infty} r(n, \theta) < \log \frac{\delta_{4,3} \delta_{2,2} \delta_{4,4}^{2k}}{\delta_{4,2} \delta_{2,1} \delta_{2,1}^{2k-1} \delta_{1,1}^{2k}} - \frac{\delta_{2,1}^{2k-1} \delta_{1,1}^{2k}}{\delta_{4,4}^{2k}} \mathrm{KL}(q_M(O|\theta_d) || q_M(O|\theta)) + \delta. \quad (48)$$

Through these results, we can further see the probability of LLMs to infer an unrelated intention :

$$p_M(\theta | S^n, x_t) = \frac{p_M(S^n, x_t | \theta) p_M(\theta)}{p_M(S^n, x_t)} \quad (49)$$

$$= e^{n \cdot r(n, \theta)} p_M(\theta) \frac{p(S^n, x_t | \theta_g)}{p_M(S^n, x_t)} \quad (50)$$

$$= e^{n \cdot r(n, \theta)} p_M(\theta) \frac{p(S^n(\theta_d), x_t)}{p_M(S^n(\theta_d), x_t)} \frac{p(\theta_g | S^n(\theta_d), x_t)}{p(\theta_g)} \quad (51)$$

$$< c_1 c_2(n) \cdot p_M(\theta) e^{n \cdot \left( \log \frac{\delta_{4,3} \delta_{2,2} \delta_{4,4}^{2k}}{\delta_{4,2} \delta_{2,1} \delta_{2,1}^{2k-1} \delta_{1,1}^{2k}} + \delta - \frac{\delta_{2,1}^{2k-1} \delta_{1,1}^{2k}}{\delta_{4,4}^{2k}} \mathrm{KL}(q_M(O|\theta_d) || q_M(O|\theta)) \right)}, \quad (52)$$

where $c_1 \triangleq \frac{p(S^n(\theta_d), x_t)}{p_M(S^n(\theta_d), x_t)} \frac{p(\theta_g | S^n(\theta_d), x_t)}{p(\theta_g)}$ denotes the ratio of priors of prompt and ground-truth intention, $c_2(n) \triangleq \frac{\delta_{4,4}^k}{\delta_{1,2} \delta_{2,1}^{k-1} \delta_{1,1}^k} e^{n\Delta(n)}$. As the $n\Delta(n)$ approaches 0, we know that $c_2$ has an upper bound C. Then we write $\log \frac{\delta_{4,3} \delta_{2,2} \delta_{4,4}^{2k}}{\delta_{4,2} \delta_{2,1} \delta_{2,1}^{2k-1} \delta_{1,1}^{2k}} + \delta - \frac{\delta_{2,1}^{2k-1} \delta_{1,1}^{2k}}{\delta_{4,4}^{2k}} \mathrm{KL}(q_M(O|\theta_d) || q_M(O|\theta))$ as $F(\theta)$, $F(\theta_d) > 0$. As the intention $\theta$ gradually moves away from the $\theta_d$, we have the function f decreasing continuously and taking a negative value at the boundary of $\Theta$. Let's take the upper bound of the function $F(\theta)$ on the boundary to be $m, m < 0$ . Because of the continuity of the function $F(\theta)$, there exists an $\epsilon$ such that $\forall \theta$ s.t. $(q_M(x, y|\theta_g) || q_M(x, y|\theta)) > \epsilon$, $F(\theta) < \frac{m}{2}$. Then the unrelated intention $\theta \notin \Theta_\epsilon$ would lead to $\lim_{n \to \infty} e^{n \cdot r(n, \theta)} \to 0$. Accordingly, LLMs will filter out unrelated intentions.

## C    PROOF OF THEOREM

To emphasize the number of examples, we use $S^n$ to represent $S^n(\theta_d)$. Then, we expand $\alpha p_M(y | S^n, x_t)$ into three terms, where $\alpha > 0$ has no effect on predicting the maximum value and thus instantiated as $\alpha = p_M(S^n, x_t)$.

$$\alpha p_M(y | S^n, x_t) = \alpha \int_\Theta p_M(y | S^n, x_t, \theta) p_M(\theta | S^n, x_t) d\theta \quad (53)$$

$$= \frac{\alpha}{p_M(S^n, x_t)} \int_\Theta p_M(y | S^n, x_t, \theta) p_M(S^n, x_t | \theta) p_M(\theta) d\theta \quad (54)$$

$$= p_M(y | S^n, x_t, \theta_g) p_M(S^n, x_t | \theta_g) p_M(\Theta_\epsilon) \quad (55)$$

$$+ \int_{\Theta \backslash \Theta_\epsilon} p_M(y | S^n, x_t, \theta) p_M(S^n, x_t | \theta) p_M(\theta) d\theta \quad (56)$$

$$+ \int_{\Theta_\epsilon} (p_M(y | S^n, x_t, \theta) p_M(S^n, x_t | \theta) - p_M(y | S^n, x_t, \theta_g) p_M(S^n, x_t | \theta_g)) p_M(\theta) d\theta. \quad (57)$$

Then, let us look at the first term in the expansion, i.e, Eq. 55, $p_M(S^n, x_t | \theta_g) p_M(\Theta_\epsilon)$ is a constant independent of $y$, so the maximum points of $p_M(y | S^n, x_t, \theta_g) p_M(S^n, x_t | \theta_g) p_M(\Theta_\epsilon)$ and $p_M(y | S^n, x_t, \theta_g)$ are the same. And in the first part, we will show that when the prediction error and the initial distribution offset of the LLMs are controlled, the maximum points of $p_M(y | S^n, x_t, \theta_g)$ and $p(y | x_t, \theta_g)$ are the same, that is, the maximum prediction of an LLM based on accurate intention and the prompt is consistent with the maximum prediction of the actual downstream task.

At the same time, the second term Eq. 56 and third term Eq. 57 are regarded as terms of prediction noise. In this context, we will give the noises an upper bound in the second part. When the noises are small enough, Theorem 1 is established: the model can accurately obtain the output corresponding to the test input through the prompt composed of demonstration and test input.

## C.1 ACCURATE PREDICTION OF GROUND-TRUTH OUTPUT

In this section, we will demonstrate that if the prediction error of LLMs and the initial distribution deviation are small enough, in-context learning through the prompt based on task intention generation leads to correct downstream task prediction. That is:

$$\arg\max_y p_M(y|S^n, x_t, \theta_g) = \arg\max_y p(y|x_t, \theta_g). \tag{58}$$

We first estimate the prediction error generated by predicting the output from the input:

$$p_M(y|S^n, x_t, \theta_g) = \prod_{i=2}^{k_t} p_M(y_t^i|Y_t^{1:i-1}, \theta_g) p_M(y_t^1|S^n, x_t, \theta_g) \tag{59}$$

Similarly,

$$p(y|S^n, x_t, \theta_g) = \prod_{i=2}^{k_t} p(y_t^i|Y_t^{1:i-1}, \theta_g) p(y_t^1|S^n, x_t, \theta_g) \tag{60}$$

Assuming that the higher order error can be controlled by the lower order error, by the polynomial expansion, we have:

$$|p_M(y|S^n, x_t, \theta_g) - p(y|S^n, x_t, \theta_g)| \tag{61}$$

$$< k_t \delta_{4,1} + \sum_{i=2}^{k_t} w_i \delta_{4,1}^i \tag{62}$$

$$< k_t \delta_{4,1} + \mathcal{O}(\delta_{4,1}), \tag{63}$$

where $w_i$ represents the coefficient corresponding to the higher-order error. When the error term does not affect the predicted maximum point, we have:

$$\arg\max_y p_M(y|S^n, x_t, \theta_g) = \arg\max_y p(y|S^n, x_t, \theta_g) \tag{64}$$

In our theoretical framework, the examples and test input are pairwise independent, that is, $p(y|S^n, x_t, \theta_g) = p(y|x_t, \theta_g)$. Therefore, we can naturally deduce the Eq 58.

However, it is important to point out that, in practice, the examples and test inputs cannot be completely independent. In this case, we can give the conditions that need to be met for predicted maximum points to be equal

Expand the task intention predictors:

$$p(y|S^n, x_t, \theta_g) = \sum_{h_t' \in \mathcal{H}} p(y|x_t, h_t', \theta_g) p(h_t'|S^n, x_t, \theta_g) \tag{65}$$

$$\propto \sum_{h_t' \in \mathcal{H}} p(y|x_t, h_t', \theta_g) p(x_t|h_t', \theta_g) p(h_t'|S^n, \theta_g) \tag{66}$$

$$= \sum_{h_t' \in \mathcal{H}} p(y|x_t, h_t', \theta_g) p(x_t|h_t', \theta_g) \sum_{h_n^f} p(h_t'|h_n^f, \theta_g) p(h_n^f|S^n, \theta_g) \triangleq WTv. \tag{67}$$

Similarly, the downstream task prediction items are as follows:

$$p(y|x_t, \theta_g) = \sum_{h_t' \in \mathcal{H}} p(y|x_t, h_t', \theta_g) p(h_t'|x_t) \tag{68}$$

$$\propto \sum_{h_t' \in \mathcal{H}} p(y|x_t, h_t', \theta_g) p(x_t|h_t', \theta_g) p(h_t') \triangleq Wu. \tag{69}$$

We use the matrix $W \in \mathcal{R}^{|\mathcal{O}|^k \times |\mathcal{H}|}$ to represent the common term $p(y|x_t, h_t', \theta_g) p(x_t|h_t', \theta_g)$ of the expansions of two distributions, the matrix $T \in \mathcal{R}^{|\mathcal{H}| \times |\mathcal{H}^f|}$ to represent the probabilistic transition matrix starting from the separator hidden state $p(h_t'|h_n^f, \theta_g)$, the vector $u \in \mathcal{R}^{|\mathcal{H}^f|}$ to represent the

probability $p(h'_t)$ and the vector $v \in \mathcal{R}^{|\mathcal{H}^f|}$ to represent the probability $p(h_n^f|S^n, \theta_g)$. And We will estimate the upper bound on the prediction probability error of the two distributions for each label $||WTv - Wu||_\infty$, and then prove that the maximum point of the two distributions is the same.

$$||WTv - Wu||_\infty \leq ||WTv - Wu||_1 \tag{70}$$

$$\leq ||W||_1 ||Tv - u||_1 \tag{71}$$

$$= ||Tv - u||_1 \tag{72}$$

Then by using $v_i$ to represent the probability $p(h_n^f(i)|S^n, \theta_g)$, we can give further contractions when the initial distribution offset is bounded:

$$||Tv - u||_1 = 2TV(q(h'_t)|| \sum_{i=1}^{|\mathcal{H}^f|} v_i q(h'_t|h^f(i))) \tag{73}$$

$$\leq 2 \sum_{i=1}^{|\mathcal{H}^f|} v_i TV(q(h'_t)||q(h'_t|h^f(i))) \tag{74}$$

$$\leq 2 \max_{h^f \in \mathcal{H}^f} TV(q(h'_t)||q(h'_t|h^f)). \tag{75}$$

If this TV distance is small enough, the distribution Eq. 67 and Eq. 69 have the same maximum points. Then the maximum points of the distribution Eq. 65 and Eq. 68 are the same. By combining Eq. 64 we achieve Eq. 58.

## C.2 EXPANSION OF THE PREDICTION TERM

Here, we provide a detailed description for expanding the prediction term, $p_M(y|S^n(\theta_d), x_t)$.

We first show that the surrogate outputs $p_M(y|S^n(\theta_d), x_t, \theta_g)$ have the same maximum points as the ground-truth outputs $p(y|x_t, \theta_g)$. This indicates that LLMs can predict the ground-truth output when conditioning on the inferred intention $\theta_g$ with bounded error $\eta_e$ that is related to the error of next-token prediction $\delta_{4,1}$,

$$\arg\max_y p_M(y|S^n(\theta_d), x_t, \theta_g) = \arg\max_y p(y|x_t, \theta_g) = y_t(\theta_g), \tag{76}$$

where the prediction of LLMs is conditioned on three terms: demonstrations $S^n(\theta_d)$, test input $x_t$, and the ground-truth intention $\theta_g$. However, in practice, the output of the in-context predictor is $p_M(y|S^n(\theta_d), x_t)$, as shown in Eq. 2. This is because the intention $\theta_g$ is a transition matrix between hidden states rather than LLMs' input[3]. Generally, these two terms are not equal, i.e.,

$$\arg\max_y p_M(y|S^n(\theta_d), x_t, \theta_g) \neq \arg\max_y p_M(y|S^n(\theta_d), x_t) = \arg\max_y \alpha p_M(y|S^n(\theta_d), x_t) \tag{77}$$

where we introduce $\alpha > 0$ having no effect on predicting the maximum value and thus is instantiated as $p_M(S^n(\theta_d), x_t)$ for simplicity. Thus, Eq. 77 implies that LLMs would fail to predict $y_t(\theta_g)$ through $p_M(y|S^n(\theta_d), x_t)$ even though the ground-truth intention $\theta_g$ has been inferred.

To bridge $p_M(y|S^n(\theta_d), x_t, \theta_g)$ the prediction with intention $\theta_g$ with model output $p_M(y|S^n(\theta_d), x_t)$, we expand model prediction into three terms as follows,

$$\alpha p_M(y|S^n(\theta_d), x_t) = p_M(S, x_t|\theta_g)p_M(\Theta_\epsilon)p_M(y|S, x_t, \theta_g)$$

$$+ \int_{\Theta \setminus \Theta_\epsilon} p_M(y|S, x_t, \theta)p_M(\theta)p_M(S, x_t|\theta)d\theta \tag{78}$$

$$+ \int_{\Theta_\epsilon} p_M(\theta)(p_M(y|S, x_t, \theta)p_M(S, x_t|\theta) - p_M(y|S, x_t, \theta_g)p_M(S, x_t|\theta_g))d\theta,$$

where $\alpha > 0$ and $S$ represents $S^n(\theta_d)$, $p_M(\Theta_\epsilon) = \int_{\Theta_\epsilon} p_M(\theta)d\theta$ captures the demonstration shift. We show that the first term of the expansion in Eq. 78 has the same maximum value as the ground truth when the prediction error $\eta_e$ in Eq. 7 is small. This suggests that the difference (termed prediction noise) is from the last two terms of expansion in Eq. 78 when the prediction error is small.

---

[3]Discussion on whether it is possible to introduce an intention embedding as input can be found in Sec. 7.

## C.3 UPPER BOUND OF THE NOISE TERMS

In this section, we will estimate the second and third terms, which are highly correlated with the ability of LLMs to distinguish intention and predict output.

The second item Eq. 56 represents the ability of LLMs to distinguish intention.

Using previous analysis, $\forall \theta \notin \Theta_\epsilon, F(\theta) < \frac{m}{2}$. Then we can estimate the item Eq. 56

$$\int_{\Theta \backslash \Theta_\epsilon} p_M(y|S^n, x_t, \theta) p_M(S^n, x_t|\theta) p_M(\theta) d\theta \tag{79}$$

$$= p(S^n, x_t|\theta_g) \int_{\Theta \backslash \Theta_\epsilon} p_M(y|S^n, x_t, \theta) e^{n \cdot r(n,\theta)} p_M(\theta) d\theta \tag{80}$$

$$< \int_{\Theta \backslash \Theta_\epsilon} e^{n \cdot r(n,\theta)} p_M(\theta) d\theta \tag{81}$$

$$< C \, e^{n \cdot \frac{m}{2}}, \tag{82}$$

where C is defined in Appendix B , as the number of examples n increases, the noise of this term tends to 0.

The third item Eq. 57 represents the mixed noise of intention inference and output prediction of LLMs.

Using the previous theoretical analysis, we have:

$$p_M(y|S^n, x_t, \theta) p_M(S^n, x_t|\theta) = p_M(y|S^n, x_t, \theta) p_M(x_t|S^n, \theta) p_M(S^n|\theta) \tag{83}$$

$$= \prod_{l=1}^{k_t} p_M(y^l|Y^{1:l-1}, \theta) \prod_{m=2}^{k_t} p_M(x_t^m|X_t^{1:m-1}, \theta) \prod_{j=1}^{n} [\, p_M(o'_{j+1}|o_j^f, \theta) \prod_{i=2}^{2k} p_M(o_i^j|O_{1:i-1}^j, \theta) \,], \tag{84}$$

where $y^0 \triangleq x_t^{k_t}, o'_{n+1} \triangleq x_t^1, o_1^j \triangleq o'_j$.

Combining assumption 3 and assumption 5, we can estimate the noise of intention bias on the model's prediction of the next token, $\forall \theta \in \Theta_\epsilon$:

$$|p_M(o_i|o_{i-1}, \theta) - p_M(o_i|o_{i-1}, \theta_g)| \tag{85}$$

$$\leq |p(o_i|o_{i-1}, \theta) - p_M(o_i|o_{i-1}, \theta)| + |p(o_i|o_{i-1}, \theta) - p(o_i|o_{i-1}, \theta_g)| + |p(o_i|o_{i-1}, \theta_g) - p_M(o_i|o_{i-1}, \theta_g)| \tag{86}$$

$$< 2\delta_{4,1} + \delta_3 \tag{87}$$

Considering polynomial expansion, assuming that the sum of higher-order term can be controlled by lower-order term, we have:

$$|\int_{\Theta_\epsilon} (p_M(y|S^n, x_t, \theta) p_M(S^n, x_t|\theta) - p_M(y|S^n, x_t, \theta_g) p_M(S^n, x_t|\theta_g)) p_M(\theta) d\theta| \tag{88}$$

$$< \int_{\Theta_\epsilon} (n\delta_5 + (n(2k-1) + 2k_t - 1)(2\delta_{4,1} + \delta_3) + \sum_{i=2}^{n+2} q_i \delta^i) p_M(\theta) d\theta \tag{89}$$

$$< (n\delta_5 + (n(2k-1) + 2k_t - 1)(2\delta_{4,1} + \delta_3) + \mathcal{O}(\delta_{4,1} + \delta_3 + \delta_5)) \, p_M(\Theta_\epsilon). \tag{90}$$

where $q_i$ represents the coefficient corresponding to the higher-order term.

Combining the above analysis, as the term 82 decreases under certain conditions, the terms 63 and 75 are acceptable offset, when the term 90 are small enough, ICL reach accurate downstream task prediction.

However, there might exist Type-II unrelated intentions such that $r(n, \theta) > 0$. In this case, the estimate above does not yield good results. Possible reasons include excessive noise caused by delimiters and the inability of the model to distinguish intention.

## D    MORE DETAILS OF INTENTION MODEL

Eq. 2 leads to two theoretical and two empirical challenges. Theoretically, it is challenging to figure out when ICL emerges under the scenario of the discrepancy between the ground-truth intention and the demonstration intention. In addition, modeling the prediction ability of LLMs is challenging under the ICL scenario. In this context, introducing a relatively strong assumption can explain some facets of ICL (Xie et al., 2022; Wang et al., 2024) while overlooking the difference in performance among LLMs. Accordingly, theoretical explanations would fail to match the empirical understanding of ICL. Empirically, it is challenging to construct demonstrations with optimal intentions, i.e., $\theta_d \neq \theta_g$, leading to the demonstration shift. This is consistent with the advanced explorations in constructing appropriate demonstrations. Meanwhile, producing expected outputs is challenging when the employed LLMs exhibit relatively poor performance, aligning with the outstanding studies in fine-tuning LLMs to promote ICL. Our intention model aims to address these theoretical challenges, providing insights to address empirical challenges by connecting theoretical explanations and empirical understanding.

## E    RELATED WORK

**Large Language Models.** Based on highly parallelizable Transformer architecture (Vaswani et al., 2017), which incorporates self-attention mechanisms, BERT (Devlin et al., 2019) emerged as a groundbreaking bidirectional language model through pre-training. This study has inspired a large number of follow-up work, establishing the "pre-training and fine-tuning" learning paradigm, e.g., GPT-2 (Radford et al., 2019) and BART (Lewis et al., 2020)). Researchers have further observed that scaling models often exhibit distinct behaviors from their smaller counterparts and display surprising capabilities. Consequently, the term "large language models (LLMs)" have gained significant traction. Typically, LLMs refer to Transformer-based language models encompassing hundreds of billions or more parameters, trained on immense text data collections(Shanahan, 2023), such as GPT-3 (Brown et al., 2020), Mistral (Jiang et al., 2023) and LLaMA (Touvron et al., 2023). LLMs have revolutionized various NLP tasks, such as text summarization, question answering, and translation, with their explosive growth in size significantly enhancing their capacity to comprehend human language (Zhao et al., 2023).

**Adapting LLMs through fine-tuning.** Similar to traditional transfer learning, fine-tuning is the straightforward approach to adapting pre-trained models to specific downstream tasks, involving adjusting the model parameters to optimize performance on a given dataset. In this regard, parameter-efficient fine-tuning has become a preferred choice, which focuses on updating only a subset of parameters (Hu et al., 2021; Houlsby et al., 2019; Ouyang et al., 2022). These methods significantly enhance LLMs' performance on downstream tasks.

**Adapting LLMs through in-context learning.** Different from traditional transfer learning, LLMs showcase a remarkable in-context learning (ICL) capability, which is a framework that empowers LLMs to grasp tasks through conditioning on some demonstrations (Dong et al., 2023). Namely, LLMs can do a downstream task by simply conditioning on a few input-output demonstrations without the need to update model parameters for novel tasks. Similar to transfer learning to search for optimal model parameters, advanced works aim to search for good demonstrations (Liu et al., 2022; Wang et al., 2024), where it shows that the label space presented, the overall format of the presentation, and the selection of demonstrations play a crucial role in ICL (Min et al., 2022). Besides practical applications, there have been some theoretical explorations on the mechanism of ICL. Xie et al. (2022) conceptualize in-context learning as a language model capable of conducting implicit Bayesian inference, theoretically proving that LLMs can identify and learn latent conceptual variables embedded within examples. Recent work points out that ICL is not conventional learning when learning label relationships (Kossen et al., 2024). Pan et al. (2023) decouple the ICL ability into task recognition ability and task learning ability, and further show how they utilize demonstrations. Wies et al. (2023) segment the ICL framework into two distinct stages: pre-training and context learning, and leverage the PAC framework to provide a theoretical analysis of ICL. The existing works make a great contribution to explaining ICL, while their connections to empirical explorations are relatively weak. Our work mainly focuses on making theoretical explanations align well with empirical understanding.

Table 2: Recognition accuracy (%) evaluated with various LLMs' architecture, different LLMs' layers, and the diverse number (32 and 64) of tokens in the outputs for constructing features.

| LLMs | Layer | Question + outputs | Question (64) | Question (32) | Question |
|---|---|---|---|---|---|
| LLaMA-2 | Low | 57.81 | 65.81 | 67.59 | 64.99 |
| | Mid | 78.25 | 87.15 | 89.08 | 90.40 |
| | High | 75.05 | 85.58 | 87.67 | 88.83 |
| Vicuna | Low | 59.00 | 64.49 | 66.48 | 67.29 |
| | Mid | 78.21 | 85.71 | 88.14 | 90.92 |
| | High | 74.40 | 82.68 | 84.77 | 87.90 |
| Mistral | Low | 47.57 | 48.59 | 47.59 | 42.62 |
| | Mid | 79.33 | 85.58 | 87.76 | 90.56 |
| | High | 73.22 | 81.24 | 84.32 | 86.38 |

# F  MORE EXPERIMENTAL DETAILS

## F.1  INTENTION RECOGNITION

The intuition is related to the basic idea of our intention model: LLMs can infer intentions from the demonstrations. This implies that intentions could be recognized. Thus, we collect prompts with distinct intentions and extract the features of these prompts using different LLMs, leading to numerous pairs of features and labels representing intentions. Subsequently, we train models on the training set and evaluate them on the test set. The results are given in Table 2, indicating that intentions can be accurately recognized. However, leveraging low-level features achieves a low prediction accuracy. We further explore the relationship between accuracy and the number of words from the response, i.e., how many words from the output are leveraged as features for intention prediction. We observe a decrease in accuracy with increased input numbers of words for all language models and layers. We find that merely using the input prompt achieves the best prediction accuracy.

**Text description.** The collection methodology involves collaboration between humans and LLMs. We generate initial prompts and group them into 2 to 5 related categories per batch to encourage LLMs to expand the prompt set, where ChatGPT 4  (OpenAI, 2023) and LLaMA-2 70B (Touvron et al., 2023) are leveraged. We employed specific strategies to guide ChatGPT in generating high-quality prompts that are challenging for LLMs to distinguish. This strategy makes verbs in the prompts vague, creating hybrid prompts. Specifically, the hybrid prompts allow the keyword "social media" to appear across various categories within the same batch, not just limited to the intention of "Social Media Content Retrieval." Moreover, the prompts in the same batch are thematically connected by similar topics, e.g., the Olympics, enhancing their contextual relevance. These strategies ensure the prompts are challenging to categorize, maintaining their intended thought without compromising distinctiveness. All prompts generated by LLMs are subject to manual screening and verification by two human annotators to ensure quality and relevance.

**Dataset description.** We collect 10, 150 samples from 50 distinct intentions to construct a novel dataset, where each intention class comprises about 200 samples. As we merely consider the scenario with non-mixed intention, each sample is devised to elicit a specific intention. In this context, we can train a model to classify LLMs' features of these samples. To ensure that the classifier recognizes intuitions based on intention patterns of features extracted by LLMs rather than specific keywords in the prompts, we avoid cases where all that is needed to infer the "translation" intention is to see the word translation. In this context, the development of the dataset should be a detailed process that integrates both linguistic and cognitive insights. Thus, the dataset is designed to accurately reflect human intuitions by analyzing various language usages and cognitive patterns (Talmy, 2019), where it categorizes LLM intuitions into facets such as retrieval, communication, creativity, and imitation, among others, providing a human-centric perspective. More details can be found in the Appendix F.1.

**Experimental setting.** In our experiments, we select three 7-billion parameter LLMs to extract features of prompts: LLaMA-2 (Touvron et al., 2023), Vicuna (Chiang et al., 2023), and Mistral (Jiang et al., 2023). These models are chosen for their robustness and diversity in architecture, allowing for a comprehensive analysis of intuitions across different systems. This selection enables

us to recognize intentions under various LLM architectures effectively. The hidden features of these models are leveraged for intention recognition, where we utilize low-level, mid-level, and high-level features to explore whether LLMs' intentions are influenced by deeper or shallower transformer blocks. For a 32-layer LLM, we define the low-level feature as originating from the 1st layer, the mid-level feature as originating from the 16th layer, and the high-level feature as originating from the last (32nd) layer. This allows us to dissect and analyze the contributions of different layers to the overall intention of the LLMs, providing insights into how various levels of abstraction within the model contribute to its final output. We also study whether the generated outputs cause changes in recognition accuracy. Each layer outputs a hidden state with a dimensionality of $4,096$. We employ a 2-layer neural network as the classifier. The network's architecture comprises two layers: the first layer has an input size of $4,096$ and an output size of $1,024$, while the second layer has an input size of $1,024$ and outputs 50 classes. We train the classifier with a standard cross-entropy loss function, Adam optimizer with the learning rate of $1e^{-3}$, and a batch size of $128$.

### F.2 INTENTION LOCATING

Besides recognizing intention, we further locate intentions in LLMs at an induction heads level, aiming to identify the instantiation of intentions. Intuitively, some specific induction heads mainly correspond to a specific intention. Therefore, we aim to identify a set of induction heads for a given intention and verify the impact of these intentions on the generation process under a specific intention. It can be seen that only a small fraction of heads are frequently involved in generation for a specific intention. To verify whether these heads are related to the same intention, we perturb these identified heads of samples, which are different from those used in the intention-locating process. For instance, replacing values of these heads with zeros (Goldowsky-Dill et al., 2023). We also randomly sample an equal number of induction heads for perturbation as a control experiment. The effect of perturbing induction heads on LLMs' performance is shown in Fig. 1. We can see that the heads identified with distinct samples can significantly reduce prediction accuracy when they are perturbed, a phenomenon not observed on randomly selected heads. Thus, these identified induction heads are highly related to the intention.

**Locating strategy.** The strategy for locating a certain intention is straightforward. We just need to compare the induction heads when the intention of interest is activated and when it is not. This is inherent to the construction of counterfactual examples through causal intervention, which has shown outstanding potential in ICL (Goldowsky-Dill et al., 2023; Madaan & Yazdanbakhsh, 2022). Specifically, we can leverage a sample to activate a certain intention while slightly modifying the sample so that it is unable to activate the intention of interest. Subsequently, we can locate the intention through the difference in induction heads under the activated and non-activated conditions. To this end, we employ the elaborate design proposed in Goldowsky-Dill et al. (2023) to realize the strategy. Specifically, given a reference sample, $x_r$, used for activating a certain intention, we construct a corresponding counterfactual example $x_c$ to deactivate the intention with minimal changes to $x_r$. Subsequently, we replace the induction heads of $x_r$ with those of $x_c$, where the heads of $x_c$ cannot activate intentions. Accordingly, we can record the output changes when replacing each head. Thus, induction heads that cause drastic changes in outputs are located as the candidate heads related to the current intention. Note that it is unclear whether $p_M(\theta|S^n(\theta_d), x_t)$ or $p_M(y|S^n(\theta_d), x_t, \theta)$ causes the change, thus, a fine-grained exploration would be promising, This is related to the exploration of internal mechanisms of LLMs (Geva et al., 2021; 2022; Belrose et al., 2023) and out-of-scope of this work.

**Experimental setting.** In our experiments, we leverage LLaMA-2-7B (Touvron et al., 2023) for intention locating. Following previous work (Wei et al., 2023), we leverage the SST-2 dataset (Socher et al., 2013) constructed for sentiment analysis and sample 100 samples from the dataset as $x_r$. Details about the construction of counterfactual examples are as follows. The reference samples are from SST-2 (Socher et al., 2013), which can activate the ICL ability. According to the rule proposed in Goldowsky-Dill et al. (2023), we slightly modify these reference samples $x_r$ to generate counterfactual samples $x_c$, while ensuring $x_c$ cannot activate the intention. Here, we give some examples in Fig 2.

$x_r$: { "text": "Input: a pale imitation \n Output: Negative \n Input: carries you along in a torrent of emotion \n Output: Positive \n Input: trashy time \n Output: Negative \n Input: all the complexity and realistic human behavior of an episode of general hospital \n Output: Negative \n Input: hold dear about cinema , \n Output: Positive \n Input: hide new secretions from the parental units \n Output: ", "answer": "Negative", "TEMPLATE_IDX": 1 }

$x_c$: { "text": "Input: a pale imitation \n Output: Nothing \n Input: carries you along in a torrent of emotion \n Output: None \n Input: trashy time \n Output: Nothing \n Input: all the complexity and realistic human behavior of an episode of general hospital \n Output: Nothing \n Input: hold dear about cinema , \n Output: None \n Input: hide new secretions from the parental units \n Output: ", "answer": "Nothing", "TEMPLATE_IDX": 1 }

Figure 2: Illustration of reference samples $x_r$ and counterfactual samples $x_c$.

