# OpenReview forum: "Intention Model: A Novel Explanation for In-context Learning"
_ICLR.cc/2025/Conference — Submitted to ICLR 2025_

### Official Review · Reviewer_b9hR · 2024-10-27

**Soundness:** 3
**Presentation:** 2
**Contribution:** 2
**Rating:** 6
**Confidence:** 3

**Summary:**

This paper proposes a theoretical framework called the "intention model" to explain ICL behaviors. The authors present a "no-free-lunch" theorem for ICL, showing that its emergence depends on prediction error, prediction noise, the model's smoothness, and demonstration quality. Unlike previous approaches with strong assumptions, this work relaxes the assumptions on perfect model alignment and demonstration representation. The intention model helps bridge the gap between theoretical explanations and empirical observations.

**Strengths:**

1. The theoretical analysis breaks away from the typical assumptions about perfect model alignment. It feels like it’s providing a more grounded explanation, making it easier to connect theory with the real behaviors of LLMs.

2. The writing is generally clear, and the mathematical notation is thoroughly defined, which makes it easier for readers to follow.

**Weaknesses:**

1. The paper's theoretical framework largely follows the derivation approach from Xie et al. (2022), particularly leveraging Bayesian Inference. Although it extends the original work by adding an error term between the LLM and the real distribution, this extension doesn’t feel groundbreaking. The contribution seems more like an incremental step rather than a major theoretical innovation.

2. The use of the term "No Free Lunch" for the presented theorem seems a bit off. The typical connotation of "No Free Lunch" is about the impossibility of a universal solution that works optimally in all scenarios. Here, the theorem implies that LLM performance depends on factors like prediction error, prediction noise, and demonstration quality. While there is indeed an implication of trade-offs, if the theorem isn’t emphasizing a broad, universal limitation but rather a specific condition for ICL, then this choice of terminology could easily confuse readers.

3. The experimental section lacks clarity on how each of the theoretical components, particularly the terms in Equation (13), manifests in practice. It’s unclear how specific terms like "error in predicting the next token," "prediction smoothness," and "distribution smoothness" are reflected in the real experimental observations. This disconnect makes it difficult for readers to see how well the theory aligns with the empirical results, and it weakens the overall support for the claims made in the theoretical part.

**Questions:**

N/A

---

> ### Author Response · Authors · 2024-11-22
> **Responses to Reviewer b9hR**
>
> We would like to express our gratitude to the reviewer for the time and effort dedicated to reviewing our work. In response to your comments, we have provided detailed responses below.
>
> ## Response to weaknesses:
>
> > **W.1**:  The paper's theoretical framework largely follows the derivation approach from Xie et al. (2022). The contribution seems more like an incremental step rather than a major theoretical innovation.
>
> ***Ans for W.1):*** We apologize for the misunderstanding. To clarify our contribution, we highlight the difference between our work and the mentioned work.
>
> - Xie et al.'s work provides an outstanding framework to explain the phenomenon of ICL [r1]. However, their work uses some strong assumptions: LLMs can exactly fit distributions and perfectly delineate task intention, under which they prove that the prediction of ICL can be asymptotically optimized as the number of examples increases.
>
> - We propose a novel theoretical framework, i.e., the intention model. This allows for common models and demonstrations and derives the no-free-lunch theorem of ICL. Namely, whether ICL emerges depends on the prediction error and prediction noise, which are determined by i) LLMs’ prediction error of the next token, ii) LLMs’ prediction smoothness, and iii) the quality of demonstrations.
>
> > **W. 2**: The use of the term "No Free Lunch" for the presented theorem seems a bit off
>
> ***Ans for W. 2):***  We apologize for the misunderstanding.
>
> The use of the term "No Free Lunch" may be confusing to readers. We chose this term to show that no demonstration works best for all problems. However, in our theorem, we are not emphasizing a universal algorithm but rather a specific condition for ICL
>  We will revise the terminology in the revised paper to make it clearer and more accurate.
>
>
> > **W. 3**:  The experimental section lacks clarity on how each of the theoretical components manifests in practice
>
> ***Ans for W. 3):*** Thanks for your valuable suggestions.
>
> Our theorem shows that the ICL capacity of an LLM emerges depends on the prediction error and the prediction noise. However, it is challenging to calculate or estimate the related values, i.e., the error of next-token prediction $\delta_3$, LLM’s prediction smoothness, and demonstration shift. To validate our theorem, we design experiments to test these factors' impact implicitly.
>
> For instance, the error of next-token prediction $\delta_3$ could be related to the LLMs' performance under general tasks. The prior error of LLMs can be regarded as the difference between the LLMs’ prediction and the actual distribution.
>
> Thus, we could conclude that the errors of GPT-4 are less than those of LLaMa-7B. Applying an external transition matrix can increase the demonstration shift, which would lead to larger prediction noise according to the prediction noise in Eq. (14). To verify these theoretical insights, we evaluate the ICL performance of different LLMs under the scenario where the transition matrix is realized by an addition operation i.e., realizing $\mathcal{T}$ by y -> (y + 1) mod 5 or by a more complicated one y -> (3y + 1) mod 5. The results shown in Table 2 are consistent with our theoretical analysis.
>
>
>
> > ***Reference***
> >
> > [r1] An explanation of in-context learning as implicit bayesian inference.

---

> ### Author Response · Authors · 2024-11-28
> **Responses to Reviewer b9hR**
>
> Dear Reviewer #b9hR,
>
> We sincerely appreciate your dedicated time and effort in reviewing our work.
>
> If you have any additional questions or issues that require further clarification, please do not hesitate to let us know. We would be more than happy to address them promptly.
>
> Thank you once again for your invaluable support and contributions to improving our work. We greatly appreciate your feedback.
>
> Best regards,
>
> Authors of #9067

---

### Official Review · Reviewer_wUqa · 2024-10-31

**Soundness:** 2
**Presentation:** 1
**Contribution:** 2
**Rating:** 5
**Confidence:** 4

**Summary:**

A theory about In-Context Learning similar to Xie et al's Bayesian Inference theory. Aims to explain some characteristics of ICL noted but not explained by prior works, for example perturbations in the label space. Aims to break down the error in predictions to interpretable quantities like LLMs' performance, quality of demonstrations.

**Strengths:**

- The theory is very similar to Xie et al's Bayesian Inference theory with some modifications like neighbourhood of an intention, etc.
- The authors provide an interpretable way to connect LLM performance on next work prediction and the quality of demonstrations to the performance on ICL tasks (under their theory), which is nice.

**Weaknesses:**

- Next token error is conditioned on \theta in assumption 4. Even if the LLM infers intention, that would be solely determined by o_1:i-1, say \theta_inferred. If the LLM is well trained, we can assume that it mostly infers the right intention and hence the condition with \delta_4,1 can be satisfied. But in cases when it fails to infer the right intention, this error may be quite large. So the assumption is strong. Moreover, there is no way to get predictions from the LLM given the same context and some different intention \theta_different, as the intention inference (if that is what happens in LLMs) is implicit and can not be disentangled. The LLM will always infer the same distribution over intentions given the same context, so I don’t understand assumption 4.

- Like Xie et al, the no free lunch theorem in this paper does not explain task learning capabilities of LLMs, on completely novel tasks (\theta not in the intention family) unrelated to pretraining text.

- The whole external mapping thing does not make much sense to me. Users do not provide an external mapping when getting the model outputs; they directly present demonstrations with this transformation. If the LLM infers this mapping, it can only be implicit. Making it a part of the original intention family. It is hard to tell if a mapping like flipped labels is present in the intention family learnt by the model during pretraining. If the mapping is randomly generated, this becomes a contradiction as it is surely not present in the pretraining corpus. The authors say that they will explore this in future work, but it is an important point that makes Xie et al’s theory and this paper’s theory inconsistent with Min et al’s results where the model is able to infer the right intention with randomly generated labels.

- Experiment section is too small and severely cut (defered to the appendix). Which model? What ICL task? It is an important part of the paper and needs to be put in the main text. Also, the evidence is circumstantial. Intervening on model activations can imply so many things related to completely different theories. How can we claim that these results imply anything specifically about the intention model? This also highlights the difference between theory and practice, as the presented theory does not elicit easily verifiable causal experiments.

- The paper is very hard to read and follow. Like
  - section 3.3, should define \delta_1,1, 1,2, 4,1, etc. What do forbidden tokens mean, what are forbidden transitions?
  - citations are placed very poorly. Sometimes before the sentence, sometimes after, sometimes unrelated; without proper usage of citet/citep.
  - [nitpicky] “Advanced works”: what is advanced, and compared to what?  “fortunately consistent”: while good to know that the authors felt relieved that the method worked, it maybe inappropriate in a technical report. Some words feel too artificially placed like “enigmatic characteristics”.
  - “These intriguing phenomena highlight the difference between traditional learning and ICL Kossen et al. (2024). These seminal explorations provide a fruitful guide for understanding and explaining ICL.”  These sentences don’t flow well. which works?

  - “a relatively weak connection to empirical investigations, potentially due to strong assumptions” [ICML 2024 paper](https://arxiv.org/abs/2310.08540) illustrates this and may be appropriately cited.

  - Line 77: “Introducing an external to modify …”, external what?
  - Line 116: definition of Sn can be confusing to read.
  - o is used for both delimiter and document tokens. confusing.
  - Line 297: \theta_g is now called inferred intention, previously it was ground truth. confusing.
  - Line 292: where does m come from? What does it mean? Unclear.
  - Table 1 is referred to as Table 2 in the text.
  - Many more ...

  Although I don't believe in reducing review scores for small clarity and writing issues, this paper seriously suffers from them and hampers the readers ability to understand the concepts conveyed. I would recommend a clear rewrite with more help from experienced co-authors.

**Questions:**

- Why is it called the no-free lunch theorem? No one expects ICL to emerge in models with high next-token prediction errors, or models to perform well on under-specified ICL tasks.

- Why do we need to have a neighborhood of intentions, which are exactly modeled; compared to Xie et al’s exact intention which may have some modeling error (as is generally the case with all ML models).

- What is the difference between Assumption 3 and 5?

- Section 5.2: Why is it difficult to estimate next-token error of LLMs? And the results of this section don't mean much. Everyone would expect the ICL performance to go down with a more complex task. This does not imply that the model is performing inference of a complex intention as presented by the theory. There is no "introduction of an external matrix T”, it is all in the theory. Where is the causal link that implies that the model is figuring out this new matrix T?

In all, I find it hard to justify this paper because the theory it presents does not make any verifiable new predictions that Xie et al did not already make, and in my opinion does not explain the previously unexplained phenomena like Min et al.

I will increase the score if the paper is clearly rewritten. I know that this is a long paper and hard to put concisely in 10 pages, but in this sort of work, the paper would greatly benefit if some of the underspecified theory (which makes it hard to understand) is moved completely to the appendix, and some more readable results like the experiments are moved to the front. The distinction between the results presented in this paper and Xie et al are unclear which could have been greatly improved in the introduction section. These are just some of my personal suggestions.

---

> ### Author Response · Authors · 2024-11-22
> **Responses to Reviewer wUqa**
>
> We would like to express our gratitude to the reviewer for the time and effort dedicated to reviewing our work. We appreciate that you find our theoretical framework generally interesting and reasonable, and the provided link between next token prediction ability and ICL is valid and interesting. In response to your valuable comments, we have provided detailed responses below. We hope that our responses have satisfactorily addressed your concerns, thereby enhancing the overall quality of our work.
>
>
> ## Response to weaknesses:
>
> > **W. 1**: The assumption 4 is strong. Moreover, there is no way to get predictions from the LLM given the same context and some different intentions \theta_different.
>
> ***Ans for W. 1):*** We apologize for the potentially confusing assumption. Accordingly, we will add the following descriptions to the revision.
>
> Assumption 4 only considers cases where the model infers a relatively correct intention, and if the model infers a wrong intention, i.e., $\theta \notin \varTheta_{\epsilon}$, we expect this probability to be small and bounded by Eq. (12). So this assumption is loose.
>
> Under our theoretical framework, task demonstrations are generated based on a given task intention. It is a tiny probability event to generate the same demonstration under very different intentions, so the model will have different tendencies for the intention distribution $p_M\left(y|S^n\left(\theta_d\right), x_{t},\theta_g\right)$ generated by demonstrations generated by different intentions.
>
> > **W. 2**: The no-free lunch theorem in this paper does not explain the task learning capabilities of LLMs on completely novel tasks (\theta not in the intention family) unrelated to the pretraining text.
>
> ***Ans for W. 2):*** Thanks for your valuable comments. Trained on large amounts of data, LLMs generate amazing emergence capabilities, allowing the model to handle previously unseen tasks. We will discuss potential limitations and future work to address novel tasks.
>
>
> > **W. 3**: The whole external mapping thing does not make much sense to me.  Users do not provide an external mapping when getting the model outputs; they directly present demonstrations with this transformation.  If the LLM infers this mapping, it can only be implicit.  Making it a part of the original intention family. It is hard to tell if a mapping like flipped labels is present in the intention family learnt by the model during pretraining.  If the mapping is randomly generated, this becomes a contradiction as it is indeed not present in the pretraining corpus.
>
> ***Ans for W. 3):*** Thanks for your valuable comments. We present this external map to show more clearly how difficult it is for the model to infer different tasks. In the multiple choice test, the flipped label can be expressed by the transition matrix, which weakens the model's ability to distinguish task intention. The external mapping matters because different mapping makes a difference in the task's difficulty. e.g. The model performed worse on task $y->3y+1(\text{mod}5)$ than on task $y->y+1(\text{mod} 5)$

---

> ### Author Response · Authors · 2024-11-22
> **Responses to Reviewer wUqa**
>
> > **W. 4**: Experiment section is too small and severely cut (defered to the appendix). Also, the evidence is circumstantial.
>
> ***Ans for W. 4):*** Thanks for your valuable comments. We will revise the paper to include more details and results in the main text. Accordingly, we will add the following descriptions to the revision.
>
> Our intention model shows that learning behaviors, e.g., learning with flipping labels, can be modeled by multiplying a transition matrix $\mathcal{T}$ by the original transition matrix $\theta$, as shown in Eq. (14), $y_{t}(\mathcal{T}\theta_g) =  {\rm \mathop{arg\; max}\limits_{y}}\; p(y| x_{t},\mathcal{T}\theta_g), \text{with} \  y(\mathcal{T}\theta_d) = {\rm \mathop{arg\; max}\limits_{y}}\; p(y| x,\mathcal{T}\theta_d)$. Based on our intention model, we have three conclusions:
>
> - _Conclusion 1_: **introducing an external $\mathcal{T}$** to modify original outputs makes ICL more challenging. This is because the transition matrix $\mathcal{T}$ would lead to $KL(p(x,y|\mathcal{T} \theta_d) || p_M(x,y|\mathcal{T}\theta_g))=\epsilon^\prime \geq \epsilon=KL(p(x,y|\theta_d) || p_M(x,y|\theta_g))$. Consequently, this results in 1) larger prediction errors, as shown in Eq. (7); and 2) larger prediction noise, as shown in Eq. (13). Thus, introducing an external matrix $\mathcal{T}$ will degrade ICL performance, which is consistent with the results shown in **Table 2**, i.e., changing $y$ to $(y + 1)\ mod \ 5$.
>
> - _Conclusion 2_: an LLM with smaller error of next-token prediction $\delta_3$ performs **better in overriding semantic priors under flipped label scenarios**. This is because smaller prediction error $\delta_3$ can reduce the prediction noise as shown in Eq. (13). Intuitively, $\delta_3$ is related to the LLMs' performance under general tasks, i.e., $\delta_3$ of GPT4 could be less than that of GPT2. Thus, we employ three LLMs to verify the point and report their performance in **Table 2**. Our results verify the theoretical insights, and fortunately, this conclusion aligns well with the experimental observations [r1].
>
> - _Conclusion 3_: **increasing the number of demonstrations $n$ under the random label scenario lead to decreasing performance**. This is because larger $n$ would magnify the impact of demonstration shift and the LLMs’ error of next-token prediction, as shown in Eq. (13). This conclusion aligns well with the experimental observations [r3]. Similarly, a small $n$ leads to good performance, which aligns with the experimental observations [r2].
>
>
>
> > **W. 5**: The paper is very hard to read and follow.
>
> ***Ans for W. 5):*** We apologize for the clarity and writing issues in the paper. We will revise it to make it easier to read and follow and address all of the specific issues you pointed out.
>
> > **W. 6**:  Presentation can be improved by removing some theoretical proofs by putting it into the appendix and describe a stronger link between the theoretical and empirical results.
>
> ***Ans for W. 6):*** Thanks for your kind suggestion. We will put some theoretical proof into the appendix to improve the presentation. For instance, we will put Eqs. (10) and (12) in the appendix. We will add the corresponding experiments after the theoretical analysis to highlight the connection between the theoretical analysis and experiments.

---

> ### Author Response · Authors · 2024-11-22
> **Responses to Reviewer wUqa**
>
> ## Response to questions:
>
> > **Q. 1**: Why is it called the no-free lunch theorem?
>
> ***Ans for Q. 1):***
>
> The no-free lunch theorem in machine learning states that no learning algorithm is naturally met and universally superior to all others for all possible problems. In the context of this paper, the no-free lunch theorem refers to the fact that the ICL performance of an LLM will depend on the specific task and the instructions provided to the model.
>
> > **Q. 2**: Why do we need to have a neighborhood of intentions, which are exactly modeled, compared to Xie et al.’s exact intention, which may have some modeling error?
>
> ***Ans for Q. 2):***
>
> We propose a framework that models a neighborhood of intentions rather than a single exact intention. In practice, a user's true intention may not be perfectly specified, and there may be multiple plausible interpretations of the user's input. By modeling a neighborhood of intentions, we can capture a range of possible interpretations and make predictions that are robust to variations in the user's input.
>
>
> > **Q. 3**: What is the difference between Assumption 3 and 5?
>
> ***Ans for Q. 3):*** Assumption 3 describes that the difference in probability of similar intention producing the same next-token prediction in the real task space is bounded. Assumption 5 characterizes that the probability difference of similar intention in the distribution predicted by the model produces the exact next-token prediction is bounded. Actually, a significant improvement in our work is to incorporate the model's predictive power into the factors influencing ICL.
>
> > **Q. 4**: Why is it difficult to estimate next-token error of LLMs? Where is the causal link that implies that the model is figuring out this new matrix T?
>
> ***Ans for Q. 4):*** Thanks for your valuable comments. We present this external map to mainly show how difficult it is for the model to infer different tasks. In the multiple choice test, flipped label can just be expressed by the transition matrix, which will lead to the weakening of the model's ability to distinguish task intention, i.e., $KL(p(x,y|\mathcal{T} \theta_d) || p_M(x,y|\mathcal{T}\theta_g))  \geq  KL(p(x,y|\theta_d) || p_M(x,y|\theta_g))$, implicitly leads to an increase in $p_M(\varTheta_{\epsilon})$, and thus the model's performance under the flipped-label task will deteriorate.
>
>
>
> > ***Reference***
> >
> > [r1] Sang Michael Xie, Aditi Raghunathan, Percy Liang, and Tengyu Ma. An explanation of in-context learning as implicit bayesian inference.
>
> > [r2] Larger language models do in-context learning differently.
>
> > [r3]   In-context learning learns label relationships but is not conventional learning.

---

> > ### Comment · Reviewer_wUqa · 2024-11-25
> >
> > I appreciate the detailed response to my concerns. However, my core concerns still remain:
> > 1. The paper is still hard to read and follow. It would benefit from some restructuring.
> > 2. Lack of significant improvement in theory over Xie et al. The presented intention model does not explain any currently unexplained phenomena (like Min et al.). What intention is being inferred with random labelling instead of the explained flipped labelling? What transition matrix would correspond to that? The problem is that [task recall](https://arxiv.org/abs/2305.09731) even after random labelling can not be explained with "intentions" learned during pre-training. Although the intention model might be able to explain meta learning in small transformers with toy data, it is not convincing for real ICL in LLMs.
> >
> > Hence, I will keep my score.

---

> > > ### Author Response · Authors · 2024-11-28
> > > **Responses to Reviewer wUqa**
> > >
> > > Dear Reviewer #wUqa,
> > >
> > > Thank you for your invaluable feedback, which provides essential opportunities to clarify our contributions. We greatly appreciate your insights and would like to address your core concerns below.
> > >
> > > > **W.1**: The paper is still hard to read and follow. It would benefit from some restructuring.
> > >
> > > **Response**:
> > >
> > > In response to your suggestion, we have moved several equations to the appendix to enhance the clarity of the main text. The current structure is as follows, and we welcome any further suggestions you may have:
> > >
> > > - _Introduction_: We explain what ICL is and highlight the difference between ICL and traditional learning by reviewing significant works. Then, We introduce various empirical observations and studies, discuss some theoretical explanations, and outline our main contributions: **First, we propose a no-free-lunch Theorem for ICL, demonstrating that the conditions for ICL emergence are not naturally met. Second, our intention model bridges the gap between theoretical and empirical results by providing a novel explanation for ICL behavior when LLMs are conditioned on varying input-output mappings**. We also acknowledge the contributions of [ref1], while clarifying that these two points are not addressed in [ref1].
> > >
> > > - _Preliminary_: We give the defination of ICL and detail some basic concepts used in ICL.
> > >
> > >
> > > - _Setup of Intention Model_: We introduce all assumptions used in our work.
> > >
> > > - _Explaining In-Context Learning with Intention Model_:
> > >
> > > - - Section 4.1: We capture the target to explain ICL and propose to model the discrepancy between the predicted and ground-truth distributions. We find that ICL emerges by producing the ground-truth output using the ground-truth intention and inferring the ground-truth intention by reducing the probability of incorrect intentions.
> > >
> > > - - Section 4.2: We analyze the intention and output errors, leading to our no-free-lunch theorem for ICL.
> > >
> > > - - Section 4.3: We explain the learning behavior of ICL under flipped label and random label scenarios.
> > >
> > >
> > > - _Experiments_: We illustrate the intention model by locating task-related intentions and validate our theorem through a flipped label scenario.
> > >
> > > - Finally, we review related works in Sec. 6, list some limitations in Sec. 7, and make a brief conclusion in Sec. 8.
> > >
> > > We greatly respect your expertise in writing and structuring, and we would highly appreciate any further invaluable suggestions.
> > >
> > > > **W.2**: Lack of significant improvement in theory over Xie et al.
> > >
> > > **Response:**
> > >
> > > We apologize for any misunderstanding. Please allow us to clarify our contributions in comparison to [ref1].
> > >
> > > - [ref1] employs relatively strong assumptions: i) LLMs can exactly fit the training distribution, focusing mainly on the discrepancy between training and testing distributions, and ii) demonstrations perfectly delineate task intention, which may be challenging in studying flipped label scenarios. Furthermore, their results suggest that ICL predictions can be asymptotically optimized as the number of examples increases.
> > >
> > > - In contrast, we propose a novel theoretical framework, the intention model, which accommodates common LLMs and demonstrations. Our no-free-lunch theorem indicates that the emergence of ICL depends on the prediction error and prediction noise, determined by i) LLMs’ prediction error for the next token, ii) LLMs’ prediction smoothness, and iii) the quality of demonstrations.
> > >
> > > We respect and appreciate the contributions of [ref1], and we have highlighted our inspiration from their work. However, the above clarifications support our unique contributions and significant theoretical improvements.
> > >
> > >
> > > > **Q.1**: What intention is being inferred with random labeling instead of the explained flipped labeling? What transition matrix would correspond to that? The problem is that task recall even after random labelling can not be explained with "intentions" learned during pre-training. Although the intention model might be able to explain meta learning in small transformers with toy data, it is not convincing for real ICL in LLMs.
> > >
> > > **Response:**
> > >
> > > Thank you for your comments. As stated in our work, analyzing a random mechanism is more challenging because each demonstration would be generated with a distinct intention. This relates to mixed-intention scenarios, where even humans may struggle to infer the correct intentions due to random labels. We plan to explore this challenging scenario in future work.
> > >
> > > [ref1] An explanation of in-context learning as implicit bayesian inference. Xie et al.
> > >
> > > Best regards,
> > >
> > > Authors of #9067

---

> > > ### Author Response · Authors · 2024-12-03
> > > **Window for discussion is closing**
> > >
> > > Dear Reviewer wUqa,
> > >
> > > We sincerely appreciate the time you have taken to review our work and provide insightful comments. We understand that your schedule is demanding. However, as the window for discussion is closing, we kindly request that you review our responses to ensure your concerns have been addressed. We would greatly value your further feedback and are committed to making any necessary improvements to enhance our work.
> > >
> > > Best regards,
> > >
> > > Authors of 9067

---

> > > > ### Comment · Reviewer_wUqa · 2024-12-03
> > > >
> > > > I appreciate the authors' efforts during the rebuttal period. And I believe there is merit in the intention model theory but it needs to be consistent.
> > > >
> > > > It is true that inferring the transition matrix for random labelling will be challenging under this intention model, but Min et al actually showed that models can easily figure out this case where the labels are not useful and maintain performance. Pan et al then showed that this ability (called task recall) does not depend on pairing of inputs and outputs, is present in smaller models and exhibited when the number of samples are small. Larger models and more demos start to learn (task learning) this newly demonstrated mapping (even if it is random).
> > > >
> > > > Under the intention model, task recall can not be explained properly. We should always have a high error with random mapping under the intention model as the transition matrix should be hard to figure out. As the data shows this is not the case, and is different from the theory, there is something missing in the theory.
> > > >
> > > > I believe that the theory will benefit from another revision taking care of this aspect so that there are no inconsistencies between model behavior and the theory.
> > > >
> > > > I am increasing the score to 5 to reflect the potential usefulness of this theory, but will still reject it due to this inconsistency.

---

> > > > > ### Author Response · Authors · 2024-12-04
> > > > > **Responses to Reviewer wUqa**
> > > > >
> > > > > Dear Reviewer wUqa,
> > > > >
> > > > > We would like to express our sincere gratitude for your invaluable assistance during this busy period. Your in‐depth comments and suggestions have significantly enhanced the quality of our work. For instance, our paper did not initially address the issue of inconsistency. In response to your insightful comments, we will incorporate the following descriptions to clarify this inconsistency in the revised paper.
> > > > >
> > > > > Our proposed intention model can elucidate the paradoxical views on whether ICL predictions depend on in‐context labels.
> > > > >
> > > > > - Min et al. observe that leveraging randomly assigned labels for the inputs barely affects ICL performance [r1].
> > > > >
> > > > > - Conversely, Kossen et al. reject the null hypothesis that ICL predictions do not depend on the conditional label distribution of demonstrations, concluding that label randomization adversely affects ICL predictions [r2].
> > > > >
> > > > > According to our intention model theory, the random label scenario can be formulated using a random transition matrix as shown in Eq. 10. Each demonstration is assigned a distinct random transition matrix $\mathcal{T}^r$ to modify the intention of demonstrations, leading to a significant demonstration shift $\delta^r = KL(q_M(x,y|\mathcal{T}^r\theta_d)||q_M(x,y|\mathcal{T}^r\theta_d))>KL(q_M(x,y|\theta_d)||q_M(x,y|\theta_d))=\delta$. Thus, introducing random transition matrix leads to a more extensive demonstration shift. Given this background, we can address the question-- why are these contradictory results observed in [r1] and [r2]? Our intention model provides a clear explanation for this.
> > > > >
> > > > > - 1) _The demonstration shift matters_. According to our intention model theory, the output error is decomposed into three terms, $\eta_e$, $\eta_{n_1}$, and $\eta_{n_2}$. The first term $\eta_e<k_t\delta_{4,1}=\delta_{4,1}$ depends on LLMs' error of next-token prediction. Here, $k_t = 1$ as all considered datasets are classification and multi-choice tasks. The second term decreases exponentially as the number of demonstrations increases. The demonstration shift scales the last term  $p_M(\varTheta_{\epsilon})$. Thus, a larger demonstration shift results in more significant output errors.
> > > > >
> > > > > - 2) _The number of demonstration matters_. According to the prediction noise $\eta_{n_2}$ (Eq. 9), more demonstrations lead to higher prediction noise, i.e., worse predicted outcomes, given the same demonstration shift. Thus, increasing the number of demonstrations leads to worse ICL performance. In this regard, [r1] used merely 16 demonstrations, while [r2] leveraged more than 50 demonstrations to evaluate ICL performance. Consequently, [r1] observed relatively small prediction errors, while [r2] observed higher ones.
> > > > >
> > > > > Thank you once again for your in-depth comments, which have significantly improved the quality of our work. We are so excited to discuss ICL further with you.
> > > > >
> > > > > [r1] Rethinking the role of demonstrations: What makes in-context learning work? Min et al. EMNLP, 2022
> > > > >
> > > > > [r2] In-context learning learns label relationships but is not conventional learning. Kossen et al. ICLR, 2024

---

### Official Review · Reviewer_6cPC · 2024-11-03

**Soundness:** 3
**Presentation:** 1
**Contribution:** 2
**Rating:** 6
**Confidence:** 3

**Summary:**

This paper proposes a latent variable model to study in-context learning (ICL) of large language models (LLM). It contributes a theoretical framework based on hidden “intentions” which are sampled during LLM generation. A hidden Markov model is used to model the overall generation process of tokens from an intent. The paper proves a novel no free-lunch theorem for the intent model which describes the conditions for when ICL emerges in an LLM (small next-token prediction error and prediction noise). In addition, the paper relates the selection of demonstrations for ICL to matching the original intent, and provides theoretical insights for the process. Empirically, the paper reports experiments on the ability of LLMs to adapt to randomized labels in-context, linear probing for intents, and identifying induction heads for intent inference.

**Strengths:**

This work introduces a latent variable “intent” based model for understanding ICL. The model is a reasonable plausible model for ICL in LLMs, outlining the weak assumption used in the theoretical analysis. Based on the intent model, several theoretical results are given, including conditions for when ICL can emerge in LLMs, under the intent model. The model also provides theoretical understanding  to explain the phenomenon of demonstration selection for ICL, and adapting to random label changes (or other task shifts) using ICL.

The paper provides some experimental confirmation of their intent model and theoretical analysis. The experiments show the performance of LLMs under task shifts which appear to support the analysis. Moreover, experiments that use (2-layer) probes to classify intents and isolation inductions heads for intents are included, which provide some justification for their model.

The idea of changing the value of isolated induction heads for intent and observing its effect on the LLM is interesting. The results appear to confirm the importance of the identified heads.

**Weaknesses:**

The amount of detailed mathematical analysis in Sections 3 and 4 is dense and obscures the key take away messages from the theory. For example, after detailing the many assumptions for the intent model and deriving the no-lunch theorem in 4.2, the conclusion of the theorem appears to be “LLMs with weak capability in prediction the next token and non-qualified (?) demonstrations fail to exhibit ICL capability.” This is very well-known from empirical evidence (since GPT4), so it is not very surprising that the intent model, under reasonable assumptions, arrived at this result. As a key result of this paper, its relevance to a broader ICLR audience is unclear. One suggestion to the authors is to reconsider whether to keep all of the technical details in the main paper, or describe the main takeaways and the theorem, but move the rest into the appendix.

The paper lacks direct empirical confirmation of its theoretical findings. In 5.2 it states that “it is challenging to calculate or estimate these values (values predicted as necessary for ICL in Theorem 1)” hence indirect experiments must be done. This is a significant weakness for the theory, as it essentially cannot be experimentally confirmed or falsified. Can the values be estimated in a toy setting?

The intent recognition experiment is not totally convincing. It sets up an intent prediction task and uses features extracts from different layers of LLMs’, along with a 2-layer network to predict intent. Can this task be solved without using an intent model? Please consider including a baseline that plausibly does not implement an intent model. Details of the task setup are also missing. For example, what are some of the 50 intents? Are they instructions or tasks? How are train/test splits done?

A lot of the content in the appendix is highly What is “n” in Table 1?
How does Table 1 “show that larger LLMs can capture the intention”? Isn’t the result just scaling?
L1182 “group them into 2 to 5 categories” which ones? Can you provide more details or samples for the dataset preparation?
F.2 do the induction heads identified here affect intent recognition in section F.1? I.e, if you “knock out” the heads then extract features, does the intent prediction performance degrade?

relevant to the paper. For example, Appendix D which discusses the theoretical and empirical challenges. Moreover, the experiments that actually try to confirm the plausibility of the intent model within real LLMs are in Appendix F. Please discuss these experiments in the main body of the paper, state their conclusions and how they support the theory.

Writing of the paper needs significant editing and proofreading.
Just a few examples:
L076 “Introducing an external to modify” external what?
L225 “error of next-token predicting”
L375 “It shows that ICL” what is “it”?
L385 “can be wrapped in the way” what does “wrapped” mean?
L498 “GTP-4”, “achieves exciting performance” what does “exciting” mean?
L501 “matrixes”

**Questions:**

What is “n” in Table 1?
How does Table 1 “show that larger LLMs can capture the intention”? Isn’t the result just scaling?
L1182 “group them into 2 to 5 categories” which ones? Can you provide more details or samples for the dataset preparation?
F.2 do the induction heads identified here affect intent recognition in section F.1? I.e, if you “knock out” the heads then extract features, does the intent prediction performance degrade?

---

> ### Author Response · Authors · 2024-11-22
> **Responses to Reviewer 6cPC**
>
> We extend our heartfelt thanks to the reviewer for the time and effort invested in evaluating our submission. It is gratifying to learn that you acknowledge the novelty of our perspective on explaining ICL. Given your constructive comments, we have prepared comprehensive responses below to address each point raised.
>
>
> ## Response to weaknesses:
>
> > **W. 1**: The amount of detailed mathematical analysis in Sections 3 and 4 is dense and obscures the key take away messages from the theory. One suggestion to the authors is to reconsider whether to keep all of the technical details in the main paper, or describe the main takeaways and the theorem, but move the rest into the appendix
>
> ***Ans for W. 1):*** Thanks for pointing out this potentially confusing manner of writing.
> There are many empirical studies on ICL phenomena, but few theoretical studies on the actual mechanism and influencing factors of ICL exist. Our study provides a possible theoretical framework and can be used to explain some of the ICL phenomena.
> Accordingly, we will add detailed descriptions to our revision. And we will reconsider the placement of some mathematical analysis and consider moving some of them to the appendix to focus more on the main takeaways and theorem in the main paper.
>
>
> > **W. 2**: The paper lacks direct empirical confirmation of its theoretical findings.
>
> ***Ans for W. 2):*** Thanks for pointing out this potentially confusing experimental configuration. Accordingly, we will add the following descriptions to the revision.
>
> Our intention model shows that learning behaviors, e.g., learning with flipping labels, can be modeled by multiplying a transition matrix $\mathcal{T}$ by the original transition matrix $\theta$, as shown in Eq. (14), $y_{t}(\mathcal{T}\theta_g) =  {\rm \mathop{arg\; max}\limits_{y}}\; p(y| x_{t},\mathcal{T}\theta_g), \text{with} \  y(\mathcal{T}\theta_d) = {\rm \mathop{arg\; max}\limits_{y}}\; p(y| x,\mathcal{T}\theta_d)$. Based on our intention model, we have three conclusions:
>
> - _Conclusion 1_: **introducing an external $\mathcal{T}$** to modify original outputs makes ICL more challenging. This is because the transition matrix $\mathcal{T}$ would lead to $KL(p(x,y|\mathcal{T} \theta_d) || p_M(x,y|\mathcal{T}\theta_g))=\epsilon^\prime \geq \epsilon=KL(p(x,y|\theta_d) || p_M(x,y|\theta_g))$. Consequently, this results in 1) larger prediction errors as shown in Eq. (7); and 2) larger prediction noise as shown in Eq. (13). Thus, introducing an external matrix $\mathcal{T}$ will degrade ICL performance, which is consistent with the results shown in **Table 2**, i.e., changing $y$ to $(y + 1)\ mod \ 5$.
>
> - _Conclusion 2_: an LLM with smaller error of next-token prediction $\delta_3$ performs **better in overriding semantic priors under flipped label scenarios**. This is because smaller prediction error $\delta_3$ can reduce the prediction noise as shown in Eq. (13). Intuitively, $\delta_3$ is related to the LLMs' performance under general tasks, i.e., $\delta_3$ of GPT4 could be less than that of GPT2. Thus, we employ three LLMs to verify the point and report their performance in **Table 2**. Our results verify the theoretical insights, and fortunately, this conclusion aligns well with the experimental observations [1].
>
> - _Conclusion 3_: **increasing the number of demonstrations $n$ under the random label scenario lead to decreasing performance**. This is because larger $n$ would magnify the impact of demonstration shift and the LLMs’ error of next-token prediction, as shown in Eq. (13). This conclusion aligns well with the experimental observations [3]. Similarly, a small $n$ leads to good performance, which aligns with the experimental observations [2].

---

> ### Author Response · Authors · 2024-11-22
> **Responses to Reviewer 6cPC**
>
> > **W. 3**: The intent recognition experiment is not totally convincing. Details of the task setup are also missing.
>
> ***Ans for W. 3):*** Thanks for pointing out this potentially confusing configuration description.
>
> Our experiments are mainly inspired by the observation that induction heads are shown to be closely related to general ICL in LLMs [r1]. Thus, we aim to identify a set of induction heads for a given intention and verify the impact of these intentions on the generation process under a specific intention. Due to limited space, we defer the detailed setup to Appendix F. In response to your valuable comments, we will highlight these details on the main page as follows.
>
> For the algorithm to locate induction heads, we draw inspiration from the work [r2]. Namely, we construct counterfactual examples to compare the induction heads when the intention of interest is activated and when it is not. Specifically, given a reference sample used for activating a certain intention, we construct a corresponding counterfactual example to deactivate the intention with minimal changes to the reference sample. Subsequently, we replace the induction heads of the reference sample with those of the counterfactual example. Thus, we can record the output changes when replacing each head. Consequently, induction heads that cause drastic changes in outputs are located as the candidate heads related to the current intention. Following previous work, we employ the model LLaMA-2-7B and the dataset SST-2 for the employed models and datasets.
>
> We will highlight these details on the main page in response to your valuable comments.
>
>
> > **W. 4**: Please discuss these experiments in the main body of the paper, state their conclusions and how they support the theory.
>
> ***Ans for W. 4):*** Thanks for your valuable comments. Accordingly, we will add the following descriptions to the main text. We will also provide more details on the dataset preparation.
>
>
> Our experiments are designed to assess intention recognizability, pinpoint intention instantiation, and verify theoretical insights.
>
> - Intention recognition. To make the concept of intention more vivid, we design experiments to verify the recognizability of intention. The intuition is straightforward, namely, the basic idea of our intention model is that LLMs can infer intentions from the demonstrations. This implies that intentions could be recognized when LLMs are conditioned on demonstrations. Thus, we collect $10,150$ prompts with $50$ distinct intentions and extract the features of these prompts using different LLMs, leading to numerous pairs of features and the corresponding label representing intentions. Some of these samples are used to train a classifier, and the left are used to evaluate the prediction accuracy of this classifier. The results are shown in Table 1.
>
> - Intention localization. It shows that induction heads are the mechanistic source of general ICL in LLMs [1]. This motivates us to take a step further beyond recognizing intentions, namely, we aim to identify a set of induction heads for a given intention and verify the impact of these intentions on the generation process under a specific intention. Thus, we design experiments to locate intentions with results are given in Figrue 1. These results show that we can pinpoint intention instantiation, making the concept of intention more vivid.
>
> - Insights verification. Our theorem shows that the ICL capacity of an LLM emerges depends on the prediction error and the prediction noise. However, it is challenging to calculate or estimate the related values, i.e., the error of next-token prediction $\delta_3$, LLM’s prediction smoothness, and demonstration shift. To validate our theorem, we design experiments to implicitly test the impact of these factors. For instance, the error of next-token prediction $\delta_3$ could be related to the LLMs' performance under general tasks. Thus, we could conclude that the $\delta_3$ of GPT4 is less than that of LLaMa-7B. Applying an external transition matrix can increase the demonstration shift, which would lead to larger prediction noise according to the prediction noise in Eq. (14). To verify these theoretical insights, we evaluate the ICL performance of different LLMs under the scenario where the transition matrix is realized by an addition operation, i.e., realizing $\mathcal{T}$ by y -> (y + 1) mod 5 or by a more complicated one y -> (3y + 1) mod 5. The results shown in Table 2 are consistent with our theoretical analysis.

---

> ### Author Response · Authors · 2024-11-22
> **Responses to Reviewer 6cPC**
>
> > **W. 5**: The writing of the paper needs significant editing and proofreading.
>
> ***Ans for W. 5):*** Thanks for your kind suggestions. We apologize for any writing errors or unclear phrases in the paper.  We will carefully rewrite and edit the paper to address these issues.
>
> ## Response to questions:
>
> > **Q. 1**: What is “n” in Table 1?  How does Table 1 “show that larger LLMs can capture the intention”?
>
> ***Ans for Q. 1):***  “n” refers to the number of examples in the demonstrations in Table 1.
>
> Our theorem shows the no-free-lunch nature of ICL, involving prediction error and noise. Intuitively， The **error of next-token prediction** $\delta_3$ could be related to the LLMs' performance under general tasks, i.e., $\delta_3$ of GPT-4 is less than that of GPT-2. Thus, we compare the ICL capability of different models, i.e., LLaMa-7B, Mistral-7B, and GPT-4, with results in **Table 1**, verifying that LLMs with smaller $\delta_3$ exhibit higher ICL performance.
>
>
> > **Q. 2**: Can you provide more details or samples for the dataset preparation?
>
>
> ***Ans for Q. 2):*** Details, such as the dataset description and experimental settings, are deferred to the Appendix. We are glad to provide further details.
>
>
> > **Q. 3**:  F.2 do the induction heads identified here affect intent recognition in section F.1?  I.e, if you “knock out” the heads then extract features, does the intent prediction performance degrade?
>
> ***Ans for Q. 3):*** Yes. Knocking out heads usually leads to performance degradation, especially for the heads related to the current intention. However, limited performance degradation is observed if we randomly knock out heads. This is consistent with our results shown in Figure 1.
>
>
> > ***Reference***
>
> > [r1] In-context learning and induction heads
>
> > [r2] Localizing model behavior with path patching

---

> ### Comment · Reviewer_6cPC · 2024-11-28
> **Reply to rebuttal**
>
> The reviewer would like to thank the authors for taking the effort to submit detailed responses to my comments and revising the paper. I've raise my score to reflect that some of my concerns have been address and questions clarified.

---

> > ### Author Response · Authors · 2024-11-28
> > **Responses to Reviewer 6cPC**
> >
> > Dear Reviewer #6cPC,
> >
> > Thank you for your invaluable support! We greatly appreciate your feedback and contributions to improving our work.
> >
> > Should you have any outstanding questions or require further clarification on any issues, please do not hesitate to reach out. We would be more than happy to receive your constructive comments and address or discuss them with you promptly.
> >
> > Best regards,
> >
> > Authors of #9067

---

### Official Review · Reviewer_A7mQ · 2024-11-04

**Soundness:** 3
**Presentation:** 2
**Contribution:** 2
**Rating:** 6
**Confidence:** 4

**Summary:**

The current paper attempts development of a unified theoretical model of in-context learning that can help reconcile the incoherent empirical results seen in prior literature (e.g., the effect of data to label map randomization). To achieve this, the authors explicitly model the notion of "intention", i.e., the task the user wants the model to perform on a given datapoint, and assess what conditions lead to the inferred task from the model matching the intended task. This leads to a three-part decomposition of ICL error: (i) error of next-token prediction (essentially the autoregressive training loss); (ii) smoothness of predictions (how drastically they change if context is altered); and (iii) "quality" of demonstrations. All terms have intuitively expected effects and hence reconcile past empirical results.

**Strengths:**

The prime contribution from a theoretical standpoint in this paper is introduction of the user intent as a first-class citizen in theory. This helps accommodate phenomenology around experiments where alteration of the context lead to the same outputs---if the user intent remains the same, the model is likely to produce the same output. That said, I have some apprehensions on relation to past work and general presentation / experimentation.

**Weaknesses:**

- **Relation to past work.** A crucial missing reference seems to be Lin and Lee ("Dual operating model of ICL", ICML 2024). Therein, authors define a prior over tasks the model can solve and assess the effect of context size scaling to reconcile prior empirical results. It would help if authors can delineate how their contributions differ from that of Lin and Lee.

- **Experiments.** I understand the paper is theoretically driven, but there are several papers on the theory of ICL at this point and it is unclear which theoretical framework is in fact correct. I hence encourage authors to take a prediction-centric perspective: what predictive claim does your theory offer, and can you demonstrate that said claim checks out experimentally? I am happy with an entirely synthetic experiment. The currently existing experiments suggest induction heads may be the mechanism for intention inference, but that claim is mostly speculative and is not well corroborated by the current induction head knockout experiments (by knocking out induction heads, you might be removing general ICL ability, and it is unclear if where the model is failing is inference of the correct intent).

- **General presentation.** I found the writing quite convoluted in several parts of the paper. For example, the introduction has several typos and grammatical errors, and at times has unnecessarily complicated phrasing (e.g., "Numerous outstanding works have revealed the enigmatic characteristics inherent to ICL"). The citation commands are also incorrectly used---only \citet was used, with no \citep usage. If citations are to go in parentheses, then \citep should be used.

**Questions:**

- In Section 4.2, it is mentioned that the effect of high quality demonstrations is multiplicative on ICL error; however, other terms like next-token prediction accuracy and prediction smoothness have an additive effect. Intuitively, I don't follow why this would be the case. It seems to me the first factor in equation 13 (i.e., the factor with several additive terms) is merely assessing how well the model is at pretraining and modeling the distribution at hand, and the second factor (i.e., demonstration shift) assesses how good the demonstrations are at eliciting the desired capabilities. Is this the right interpretation? If not, can you better explain why the first term has additive influence of next-token error and prediction smoothness (which I would have expected to themselves have a multiplicative effect)?

---

> ### Author Response · Authors · 2024-11-22
> **Responses to Reviewer A7mQ**
>
> We would like to express our gratitude to the reviewer for the time and effort dedicated to reviewing our work. We appreciate your encouraging remarks on the applicability of our proposed theoretical framework. In response to your valuable comments, we have provided detailed responses below. We hope that these responses can satisfactorily address your concerns.
>
> ## Response to weaknesses:
>
> > **W. 1**: A crucial missing reference seems to be Lin and Lee ("Dual operating model of ICL", ICML 2024). It would help if authors can delineate how their contributions differ from that of Lin and Lee.
>
> ***Ans for W. 1):*** Thanks for your constructive comments. Accordingly, we highlight the difference between our work and the mentioned work.
>
> - Lin et al.'s work [r1] considers a specific regression task and assesses the performance of ICL [1]. Their work focuses on context length's impact on ICL and explains two Real-World phenomena.
> - We propose a novel theoretical framework, i.e., the intention model. This allows for common models and downstream tasks and derives the no-free-lunch theorem of ICL. Namely, whether ICL emerges depends on the prediction error and prediction noise, which are determined by i) LLMs’ prediction error of the next token, ii) LLMs’ prediction smoothness, and iii) the quality of demonstrations.
>
> In response to your kind suggestions, we will add the above discussions to the revision.
>
> > **W. 2**: I hence encourage authors to take a prediction-centric perspective: what predictive claim does your theory offer, and can you demonstrate that said claim checks out experimentally?
>
> ***Ans for W. 2):***  According to your suggestions, we have highlighted the theoretical claims verified by experiments in the revision.
>
> Our theorem shows the no-free-lunch nature of ICL, involving prediction error and prediction noise. Thus, a straightforward approach to validate the theorem is to calculate the prediction error and noise. However, it is challenging to calculate or estimate these related values, i.e., the **error of next-token prediction** $\delta_3$, LLM’s **prediction smoothness**, and **demonstration shift**. Thus, providing a quantitive analysis to verify the theorem is challenging.
>
> Fortunately, some related factors can be controlled implicitly.
> - The **error of next-token prediction** $\delta_3$ could be related to the LLMs' performance under general tasks, i.e., $\delta_3$ of GPT4 is less than that of GPT2. Thus, we compare the ICL capability of different models, i.e., LLaMa-7B, Mistral-7B, and GPT-4, with results in **Table 1**, verifying that LLMs with smaller $\delta_3$ exhibit higher ICL performance.
>
> - The **demonstration shift** is captured by $\epsilon = KL(p(x,y|\theta_d) || p_M(x,y|\theta_g))$. According to Eq. (14), the demonstration shift would vary with the external transition matrix $\mathcal{T}$, leading to larger demonstration shift, i.e., $KL(p(x,y|\mathcal{T} \theta_d) || p_M(x,y|\mathcal{T}\theta_g)) = \epsilon^\prime \geq \epsilon = KL(p(x,y|\theta_d) || p_M(x,y|\theta_g))$. According to Eq. (13), applying $\mathcal{T}$ leads to larger prediction noise. Thus, we evaluate ICL performance under different matrix $\mathcal{T}$ with results in **Table 2**, where $\mathcal{T}$ is realized using different mappings, i.e., $y=y$, $y = (y+1)\ mod\ 5$, and $y = (3y+1)\ mod\ 5$. This verifies that a larger demonstration shift degrades ICL performance.
>
> Aligning with your valuable comments, we failed to provide direct experimental validation related to the LLM’s prediction smoothness. Thus, we will explore some implicit approach to reflecting the nature of LLMs in our future work.
>
> > **W. 3**: I found the writing quite convoluted in several parts of the paper.
>
> ***Ans for W. 3):*** We apologize for the convoluted writing and grammatical errors in our paper. We will carefully revise the paper to make it more straightforward and more concise and correct all citation and grammatical errors.
>
> ## Response to questions:
>
> > **Q. 1**: Can you better explain why the first term has additive influence of next-token error and prediction smoothness (which I would have expected to themselves have a multiplicative effect)?
>
> ***Ans for Q. 1):*** Following your valuable question, we have added detailed explanations of the additive influence of next-token error and prediction smoothness.
> We agree that your intuition about next-token error and prediction smoothness is correct. Within the intention neighborhood inferred by LLMs, the actual effects of these two factors accumulate gradually by multiplying, leading to a more complex error term. We want this error to be easier to understand, so we use a binomial expansion and shrink the higher-order term to a constant. This provides a more intuitive understanding of the actual effect of the error term.
>
> > ***Reference***
> >
> > [r1] Dual Operating Modes of In-Context Learning, ICML 2024

---

> > ### Comment · Reviewer_A7mQ · 2024-11-23
> > **Response**
> >
> > I appreciate the authors' response. I would have hoped for a draft update, but I'm nonetheless happy to keep my score in accordance with the original draft.

---

> > > ### Author Response · Authors · 2024-11-23
> > > **Response**
> > >
> > > We sincerely appreciate your prompt feedback and deeply understand your responsibility in not raising the score for the original draft. We are working on the new version by integrating your constructive and valuable comments.

---

### Comment · Area_Chair_rvwJ · 2024-11-24

Dear Reviewers,

This is a gentle reminder that the authors have submitted their rebuttal, and the discussion period will conclude on November 26th AoE. To ensure a constructive and meaningful discussion, we kindly ask that you review the rebuttal as soon as possible and verify if your questions and comments have been adequately addressed.

We greatly appreciate your time, effort, and thoughtful contributions to this process.

Best regards,
AC

---

### Author Response · Authors · 2024-11-25
**A revised version is uploaded**

Dear ACs and Reviewers,

We sincerely appreciate the time and effort you have invested in reviewing our work. In response to the constructive comments and valuable suggestions, we have uploaded a revised version with more detailed descriptions.

We have marked the revisions with different colors corresponding to the feedback from each reviewer:

Green for Reviewer A7mQ;

Red for Reviewer  6cPC;

Blue for Reviewer  wUqa;

Gray for Reviewer  b9hR.

---

### Meta-Review · Area_Chair_rvwJ · 2024-12-30

**Metareview:**

This paper proposes an "intention model" as a theoretical framework to explain in-context learning (ICL) capabilities in large language models. The key scientific claim is that ICL emergence depends on three factors: the model's next-token prediction error, prediction smoothness, and demonstration quality. The authors present a "no-free-lunch" theorem showing these factors determine whether ICL will emerge successfully. They also use their framework to explain empirical phenomena around how models handle flipped and random labels during ICL. The reviewers appreciated the novelty of the theoretical analysis and it's focus on interpretable terms. However, the paper has several notable weaknesses, especially the gaps in clarity, high degree of similarity to related works such as Xie et al., gaps in theory relating to inconsistency in explaining random intentions as in Min et al., and lack of ability to measure the theoretical factors in real-world experiments. Some are natural eg, next-token prediction error, while others such as prediction smoothness could also potentially benefit from related work [1]. Overall, the paper was borderline and there was no strong champion for the work, and felt a revision of the work to address the weaknesses is necessary for acceptance.

[1] https://arxiv.org/abs/2406.11233

**Additional Comments On Reviewer Discussion:**

See above.

---

### Decision · Program_Chairs · 2025-01-22

Reject